# Mitigating Translationese Bias in Multilingual LLM-as-a-Judge via Disentangled Information Bottleneck

**Hongbin Zhang** [1 2]  **Kehai Chen** [1 2]  **Xuefeng Bai** [1]  **Youcheng Pan** [2]  **Yang Xiang** [2]  **Jinpeng Wang** [3]  **Min Zhang** [1 2]

## Abstract

Large language models (LLMs) have become a standard for multilingual evaluation, yet they exhibit a severe systematic "*translationese bias*". In this paper, "translationese bias" is characterized as LLMs systematically favoring machine-translated text over human-authored references, particularly in low-resource languages. We attribute this bias to spurious correlations with (i) latent manifold alignment with English and (ii) cross-lingual predictability. To mitigate this bias, we propose DIB-JUDGE, a robust fine-tuning framework that learns a minimally sufficient, judgment-critical representation via variational information compression, while explicitly isolating spurious factors into the dedicated bias branch. Furthermore, we incorporate a cross-covariance penalty that explicitly suppresses statistical dependence between robust and bias representations, thereby encouraging effective disentanglement. Extensive evaluations on multilingual reward modeling benchmarks and a dedicated translationese bias evaluation suite demonstrate that the proposed DIBJUDGE consistently outperforms strong baselines and substantially mitigates translationese bias.

## 1. Introduction

The emergence of Large Language Models (LLMs) has revolutionized evaluation paradigms (Gu et al., 2024; Li et al., 2025; Ou et al., 2024), establishing "LLM-as-a-Judge" as a standard framework for multilingual assessment (Son et al., 2024; Pombal et al., 2025; Anugraha et al., 2025; Hada et al., 2024; Fu & Liu, 2025; Doddapaneni et al., 2025). Consequently, ensuring the accuracy and robustness

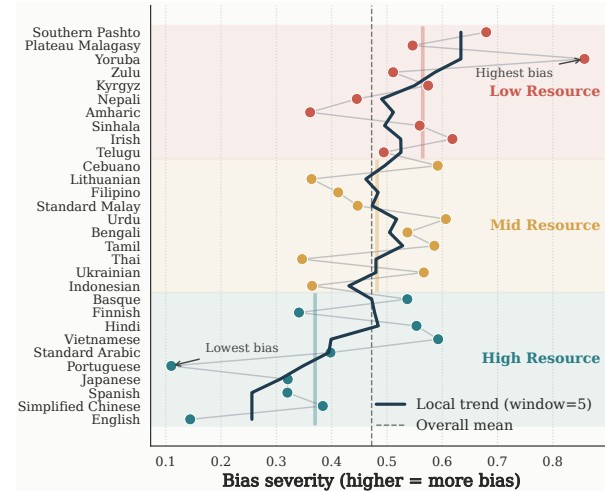

*Figure 1.* **Translationese Bias Severity of GPT-4o across languages.** Languages are sorted by resource availability from low (top) to high (bottom). The trend line illustrates the inverse relationship between resource availability and translationese bias.

of these automated judges across diverse languages has become a critical necessity (Padarha et al., 2025; Bogavelli et al., 2026).

However, the reliability of LLM judges and related LLM/MLLM evaluators is frequently undermined by systematic biases (Wang et al., 2025a; Ye et al., 2025; Gao et al., 2025; Li et al., 2024; Zhang et al., 2025d; 2026c; Zhu et al., 2025; Zhang et al., 2025b), such as position (Shi et al., 2025; Wang et al., 2024b) and verbosity bias (Saito et al., 2023). While these limitations are well-studied in English contexts (Chen et al., 2024; Zheng et al., 2023), specific failure modes within multilingual settings remain underexplored.

In this paper, we characterize a distinct bias of LLM-as-a-Judge in multilingual contexts, termed *translationese bias*, in which LLMs favor machine-translated content over human-authored reference, even when the former is semantically flawed. To investigate this bias, we first conduct a comprehensive evaluation across a diverse spectrum of languages. As shown in Figure 1, this bias is not only pervasive but is significantly exacerbated in low-resource languages. Crucially, our further attribution analysis suggests that LLM judges may conflate generation quality with two potential

[1] Institute of Computing and Intelligence, Harbin Institute of Technology, Shenzhen, China [2] Pengcheng Laboratory, Shenzhen, China [3] Keeta AI, Meituan, Beijing, China. Correspondence to: Xuefeng Bai <baixuefeng@hit.edu.cn>.

*Proceedings of the 43rd International Conference on Machine Learning*, Seoul, South Korea. PMLR 306, 2026. Copyright 2026 by the author(s).

spurious factors: (a) latent manifold alignment with English, and (b) cross-lingual predictability. While recent advancements in multilingual LLM judges have yielded promising results (Pombal et al., 2025; Anugraha et al., 2025; Zhang et al., 2026a), most existing methods remain grounded in standard Supervised Fine-Tuning (SFT). However, SFT is susceptible to exploiting spurious correlations (Shuieh et al., 2025; Gui & Ji, 2025; Chen et al., 2026), thereby limiting its efficacy in mitigating translationese bias.

To this end, we propose the Disentangled Information Bottleneck Judge (DIBJUDGE), a robust fine-tuning framework that explicitly decouples the latent representation into two components: a robust representation that preserves judgment-critical semantic information, and a bias representation that isolates the spurious factors identified above (i.e., latent manifold alignment with English and cross-lingual predictability). We leverage variational information compression to learn a robust, minimally sufficient representation that preserves only information essential for accurate judgment. To further encourage disentanglement between robust and bias representation, we penalize their mutual dependence during training. Extensive experiments on multilingual reward modeling benchmarks, including M-RewardBench (Gureja et al., 2025) and MM-Eval (Son et al., 2024), demonstrate that DIBJUDGE consistently outperforms strong baselines, yielding improved multilingual reward modeling performance. Moreover, evaluations on a dedicated translationese bias suite confirm that DIBJUDGE substantially mitigates the severity of translationese bias.

In summary, we make three key claims: (i) we characterize *translationese bias* in multilingual LLM judges and identify two related spurious factors—latent manifold alignment with English and cross-lingual predictability, (ii) we propose DIBJUDGE, a robust fine-tuning framework that disentangles judgment-critical semantics from spurious factors, and (iii) we show that DIBJUDGE consistently outperforms strong baselines on multilingual reward modeling benchmarks while effectively mitigating translationese bias.[1]

## 2. Preliminary Analysis of Translationese Bias

To systematically study translationese bias, we structure our preliminary analysis around two research questions: (i) **RQ1:** How does translationese bias vary across languages with different levels of resource availability? (ii) **RQ2:** What kinds of spurious factors are associated with this bias?

**Bias Evaluation Protocol.** We construct a controlled translationese bias benchmark derived from BELEBELE (Bandarkar et al., 2024), a multilingual reading comprehension dataset spanning 122 languages. Following the language-resource taxonomy of Joshi et al. (2020), we stratify lan-

---

[1]Our code repo is available here.

guages into high-, medium-, and low-resource tiers and sample 10 representative languages per tier. For each language, we evaluate on 200 instances, yielding a balanced benchmark across resource levels. The full list of selected languages is provided in Appendix A.1. We formulate translationese bias evaluation as a pairwise preference task, where an LLM judge compares two candidate responses for the same query: (i) a *human-authored reference* $x_H$, and (ii) a *machine-generated variant* $x_M$ obtained by standard back-translation ($L_{\text{target}} \rightarrow$ English $\rightarrow L_{\text{target}}$) to induce translationese artifacts. To avoid position bias, each instance $i$ is evaluated under both forward and reverse ordering. Details are available in Appendix A.2.

**Bias Metric Definition.** Let $y_i \in \{0, 1\}$ indicate whether the judge prefers $x_M$ in the forward order, and let $\bar{y}_i \in \{0, 1\}$ denote the corresponding preference in the reverse order. We define bias severity $\mathcal{S}_{\text{bias}}$ as the fraction of consistent judgments that favor the machine-generated output:

$$\mathcal{S}_{\text{bias}} = \frac{\sum_{i=1}^{N} \mathbb{I}\left[y_i = 1 \wedge \bar{y}_i = 1\right]}{\sum_{i=1}^{N} \mathbb{I}\left[y_i = \bar{y}_i\right]}. \tag{1}$$

### 2.1. Quantifying Bias across Resource Levels (RQ1)

Figure 1 illustrates the translationese bias severity of GPT-4o across varying language resource levels. We observe two salient patterns: first, translationese bias is pervasive across the entire linguistic spectrum; second, there is a distinct inverse correlation between resource availability and the magnitude of bias. Specifically, while high-resource languages exhibit minimal bias, low-resource languages demonstrate significantly elevated severity. These findings expose translationese bias as a severe yet previously neglected failure mode in multilingual LLM judges, which critically undermines evaluation reliability and disproportionately compromises under-resourced languages.

### 2.2. Attribution Analysis of Translationese Bias (RQ2)

Due to the scarcity of high-quality native resources (Qin et al., 2025; Huang et al., 2024), multilingual LLMs are typically pre-trained on English-dominated corpora (Kreutzer et al., 2022; Weber et al., 2024) and subsequently adapted using translated or synthetic data (Muennighoff et al., 2023; Zhang et al., 2020a; 2024b; Zuo et al., 2025; Zhang et al., 2025a). Accordingly, we hypothesize that translationese bias stems from spurious correlations induced across these two stages: (i) latent manifold alignment with English, where non-English representations are implicitly aligned to an English-centric latent space during pre-training; and (ii) cross-lingual predictability, where the judge over-relies on probability heuristics that favor the statistical patterns of machine-translated text, potentially amplified by exposure to translated or synthetic data during fine-tuning. However, causally attributing this bias to particular data mixtures

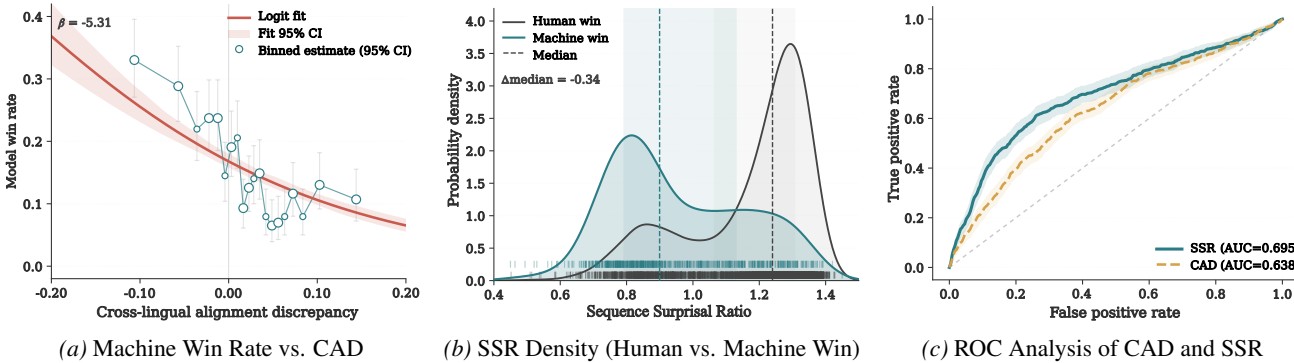

*(a)* Machine Win Rate vs. CAD     *(b)* SSR Density (Human vs. Machine Win)     *(c)* ROC Analysis of CAD and SSR

*Figure 2.* **Correlation analysis of judge preference with confounding factors.** (a) Machine win rate decreases monotonically as CAD increases, indicating that judge preference spuriously tracks latent manifold isomorphism with English. (b) SSR distributions exhibit a clear drift between human-win and machine-win cases, showing that the judge systematically favors higher-likelihood outputs. (c) ROC curves confirm that both CAD and SSR reliably predict judge outcomes, reinforcing the attribution that translationese bias is mediated by latent manifold isomorphism with English and high predicative confidence.

remains non-trivial given the opacity and heterogeneity of LLM training pipelines (Lai et al., 2025).

To address this, we introduce two measurable latent metrics that serve as quantitative proxies for these two factors: (i) *Language Alignment Score (LAS)*, defined as the degree to which a representation is geometrically aligned with an English latent manifold: $\mathrm{LAS}(x) = \frac{1}{L} \sum_{l=1}^{L} \cos\left(\mathbf{h}_l(x), \mathbf{c}_{\mathrm{en},l}\right)$, where $x$ is input sequence, $\mathbf{h}_l(x)$ is the layer-$l$ hidden representation of $x$, $\mathbf{c}_{\mathrm{en},l}$ is the English centroid at layer $l$, and $L$ is the total number of layers. (ii) *Cross-lingual Sequence Surprisal (CSS)*, defined as the length-normalized negative log-likelihood of a target sequence $x$ of $T$ tokens, conditioned on its English translated context $x_{\mathrm{en}}$: $\mathrm{CSS}(x) = -\frac{1}{T} \sum_{t=1}^{T} \log P(x_t \mid x_{\mathrm{en}}, x_{<t})$. To answer RQ2, we then investigate the extent to which the distributional divergence of these metrics between $x_H$ and $x_M$ correlates with machine win rate: (i) *Cross-lingual Alignment Discrepancy (CAD)*, $\mathrm{CAD} = \mathrm{LAS}(x_H) - \mathrm{LAS}(x_M)$, where $\mathrm{CAD} < 0$ implies that $x_M$ exhibits closer alignment to the English latent space than $x_H$ does. (ii) *Sequence Surprisal Ratio (SSR)*: $\mathrm{SSR} = \frac{\mathrm{CSS}(x_M)}{\mathrm{CSS}(x_H)}$, where $\mathrm{SSR} < 1$ indicates that $x_M$ is more cross-lingual predictable by the model relative to $x_H$.

As shown in Figure 2, LLM judges exhibit strong correlations with the introduced latent metrics. Specifically, CAD is negatively associated with machine win rate, while preferred machine outputs cluster at lower SSR, indicating a preference for English-aligned and statistically predictable sequences. ROC analysis further confirms that both metrics have meaningful discriminative power. To verify that these correlations are not merely quality artifacts, we conducted additional quality controls. COMET-Kiwi rates the human references higher in 23/29 languages ($p < 0.05$), with no significant difference in the remaining six; MQM annotation on 200 Chinese–English pairs also shows fewer errors for human text (2.58 vs. 5.41, 95% CI [2.28, 3.39],

$p = 0.0001$). Moreover, on a quality-matched six-language subset, the CAD/SSR trends persist, and a mixed-effects logistic regression with $\Delta$COMET as a covariate finds CAD ($\beta = -0.57$) and SSR ($\beta = -0.79$) significant ($p < 0.001$), while $\Delta$COMET is not ($p = 0.520$). Collectively, these results indicate that translationese bias is associated with latent manifold alignment with English and cross-lingual predictability rather than measured text quality alone.

## 3. Disentangled Information Bottleneck Judge

To mitigate the spurious correlations identified in § 2.2, we propose the Disentangled Information Bottleneck Judge (DIBJUDGE), as illustrated in Figure 3. By explicitly disentangling these spurious factors, DIBJUDGE learns a compressed representation that retains sufficient, robust features essential for accurate quality assessment.

### 3.1. Preliminaries

**Mutual Information.** Mutual Information (MI) quantifies the statistical dependence between two random variables, $A$ and $B$. Given the joint distribution $p(a, b)$ and marginals $p(a), p(b)$, MI is formally defined as :

$$I(A; B) \triangleq \mathbb{E}_{p(a,b)} \left[ \log \frac{p(a, b)}{p(a)p(b)} \right]. \quad (2)$$

**Information Bottleneck Principle.** The Information Bottleneck (IB) principle (Tishby et al., 2000; Alemi et al., 2017) seeks a compressed representation $Z$ that is *sufficient* for a target task $Y$ while remaining *minimal* with respect to the input $X$. This is formalised by minimizing the Lagrangian: $\mathcal{L}_{\mathrm{IB}} = -I(Y; Z) + \beta I(X; Z)$, where $\beta \geq 0$ governs the trade-off between prediction and compression. However, compression may discard robust semantic features in favor of simpler spurious correlations (Liu et al., 2022; Zhang et al., 2024a). Thus, relying solely on information compactness cannot guarantee the robustness.

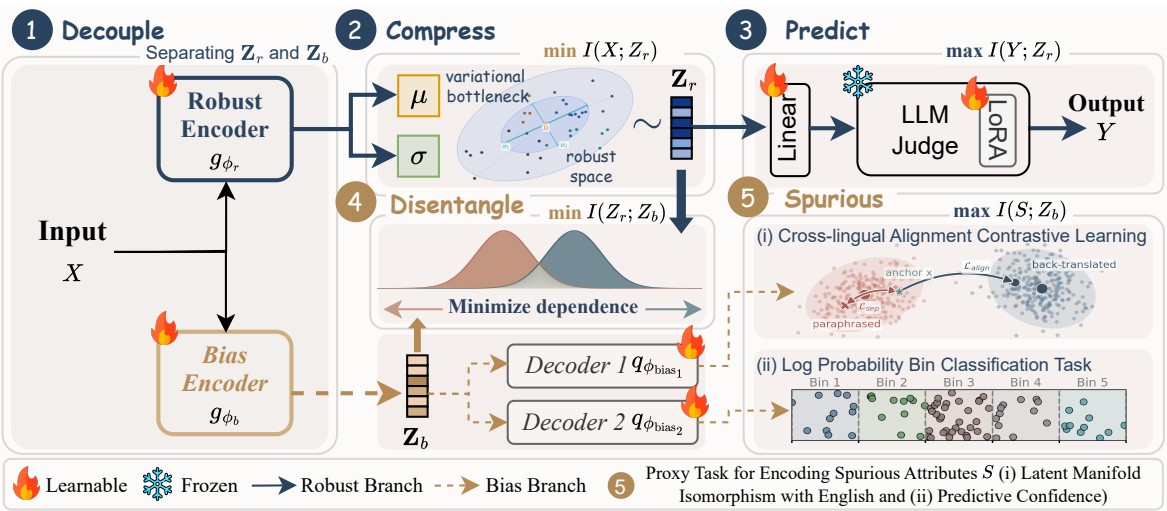

*Figure 3.* Overview of our DIBJUDGE, grounded in Equation 3. (1) It employs robust and bias encoders $g_{\phi_r}, g_{\phi_b}$ to separate $X$ into $\mathbf{Z}_r$ and $\mathbf{Z}_b$. (2) A variational bottleneck minimizes $I(X; Z_r)$. (3) The compressed $\mathbf{Z}_r$ conditions the LoRA-optimized LLM judge (Hu et al., 2022) to predict $Y$ by maximizing $I(Y; Z_r)$. (4) A cross-covariance penalty minimizes dependence $I(Z_r; Z_b)$. (5) The bias branch captures spurious attributes $S$ via two proxy tasks: cross-lingual alignment contrastive learning (CLA) and log-probability bin classification (LPBC).

## 3.2. Disentangled Information Bottleneck Objective

To prevent LLM judges from exploiting spurious shortcuts solely through information compression, we are inspired by the idea of disentangled representation learning (Wang et al., 2024d) to refine vanilla IB. Let $S$ be a spurious variable, $Z_r$ be a *relevant* variable encoding features necessary for predicting the target $Y$, and $Z_b$ be a *bias* variable serving as a dedicated "sink" to absorb $S$. We formalize the Disentangled Information Bottleneck Objective as:

$$\mathcal{L}_{\text{DIB}} = \underbrace{-I(Y;Z_r)}_{\text{Prediction}} + \underbrace{\beta\,I(X;Z_r)}_{\text{Compression}}$$
$$\quad - \underbrace{\gamma\,I(S;Z_b)}_{\text{Bias Capture}} + \underbrace{\lambda\,I(Z_r;Z_b)}_{\text{Disentanglement}}. \quad (3)$$

The first two terms apply a vanilla IB constraint restricted to the robust channel $Z_r$; the third term makes $Z_b$ informative about spurious attributes $S$; and the final term penalizes dependence between $(Z_r, Z_b)$, encouraging $Z_r$ to exclude spurious correlations that are explicitly routed into $Z_b$. Directly optimizing this objective is computationally intractable due to the difficulty of estimating mutual information in high-dimensional spaces (Liu et al., 2024b). To address this, we derive tractable variational surrogate objectives as follows:

**Decouple Robust and Bias Representation.** Using separate encoders $g_{\phi_r}$ and $g_{\phi_b}$, we decompose the input $X$ into a robust representation $\mathbf{Z}_r = g_{\phi_r}(X) \in \mathbb{R}^{T \times d}$ and a bias representation $\mathbf{Z}_b = g_{\phi_b}(X) \in \mathbb{R}^{T \times d}$, where $T$ denotes the sequence length and $d$ the feature dimension. We leverage $\mathbf{Z}_r$ for task prediction while using $\mathbf{Z}_b$ only during training.

**Compression via Variational Information Constraints.**

To facilitate compression, we leverage the Variational Information Bottleneck (Alemi et al., 2017), which imposes an upper bound on $I(X; Z_r)$ via variational inference.

**Proposition 3.1.** *Let $Z_r$ be a continuous random variable, with variational posterior $q_\phi(Z_r|X)$ and fixed prior $p(Z_r)$. Then $I(X; Z_r) \leq \mathbb{E}_{x \sim p(X)}\left[D_{\text{KL}}(q_\phi(Z_r|x)\|p(Z_r))\right]$.*

Guided by Proposition 3.1 (proved in Appendix. B.1), we can constrain $I(X; Z_r)$ via penalizing the KL divergence between the variational posterior and fixed prior. Accordingly, we adopt a standard Gaussian prior $p(Z_r) = \mathcal{N}(\mathbf{0}, \mathbf{I})$ and parameterize the variational posterior $q_\phi(\mathbf{z}_{r,t}|x)$ at each time step $t$ as a multivariate Gaussian $\mathcal{N}(\boldsymbol{\mu}_t, \boldsymbol{\sigma}_t^2)$. The resulting compression objective, defined as the average KL divergence over the sequence length $T$ and feature dimension $d$, is derived as follows (details in Appendix. B.2):

$$\mathcal{L}_{\text{compress}} = \frac{1}{T}\sum_{t=1}^{T} D_{\text{KL}}\big(\mathcal{N}(\boldsymbol{\mu}_t, \boldsymbol{\sigma}_t^2) \,\big\|\, \mathcal{N}(\mathbf{0}, \mathbf{I})\big)$$
$$= -\frac{1}{2T}\sum_{t=1}^{T}\sum_{j=1}^{d}\Big(1 + \log\sigma_{t,j}^2 - \mu_{t,j}^2 - \sigma_{t,j}^2\Big). \quad (4)$$

To allow for backpropagation, we sample the latent representation $\mathbf{z}_{r,t}$ using the reparameterization trick (Kingma et al., 2015): $\mathbf{z}_{r,t} = \boldsymbol{\mu}_t + \boldsymbol{\sigma}_t \odot \boldsymbol{\epsilon}$, where $\boldsymbol{\epsilon} \sim \mathcal{N}(\mathbf{0}, \mathbf{I})$.

**Variational Mutual Information Maximization.** The objective (Eq. 3) necessitates maximizing mutual information along two disentangled pathways: the task-predictive term $I(Y; Z_r)$ and the bias-capturing term $I(S; Z_b)$. We then maximize a variational lower bound on the mutual information guided by Proposition 3.2 (proved in Appendix. B.3).

**Proposition 3.2.** *Let $U$ and $V$ be random variables with joint distribution $p(U, V)$. For any variational conditional distribution $q_\theta(U \mid V)$, the mutual information satisfies $I(U; V) \geq \mathbb{E}_{(U,V) \sim p(U,V)}[\log q_\theta(U \mid V)] + H(U)$, where $H(U)$ denotes the marginal entropy of $U$.*

For the robust pathway, we treat the LLM judge $f_{\text{judge}}$ as the variational decoder $q_\theta$. We condition the generation on a sequence formed by concatenating the instruction embeddings $\mathbf{E}_{\text{inst}}$ with the sampled robust representation $\mathbf{Z}_r$. The task loss is defined as:

$$\mathcal{L}_{\text{task}} = \mathbb{E}_{X,Y} \left[ -\sum_{t=1}^{|Y|} \log f_{\text{judge}}(Y_t \mid [\mathbf{E}_{\text{inst}}; \mathbf{Z}_r], Y_{<t}) \right]. \quad (5)$$

We employ lightweight MLP decoders $q_{\psi_{\text{bias}}}$ and minimize the negative log-likelihood of the spurious attribute $S$ given the bias representation $\mathbf{Z}_b$ to facilitate the encoding of spurious information into the bias pathway:

$$\mathcal{L}_{\text{bias}} = \mathbb{E}_{X,S} \left[ -\log q_{\psi_{\text{bias}}}(S \mid \mathbf{Z}_b) \right]. \quad (6)$$

We operationalize the identified spurious factors through two proxy tasks that do not require gold parallel corpora. **Cross-Lingual Alignment (CLA)** targets latent-manifold alignment with English: for a non-English input, we translate it once into English and use an InfoNCE objective to align the bias representation with this translated-English manifold. **Log-Probability Bin Classification (LPBC)** targets cross-lingual predictability: we compute the base model's average token log-probability for the target sequence conditioned on its English translation, discretize the score into bins, and train $\mathbf{Z}_b$ to predict the corresponding bin. These single-forward-pass auxiliary losses are optimized jointly with the task, compression, and disentanglement objectives.

**Disentanglement via Cross-Covariance Penalty.** Directly minimizing the mutual information $I(Z_r; Z_b)$ is generally intractable. However, in high-dimensional regimes typical of LLMs, representation distributions are often well-approximated by Gaussian statistics (Lee et al., 2018; Hron et al., 2020). Under this Gaussian assumption, minimizing mutual information reduces to minimizing the cross-covariance between latent variables (Cover, 1999; Hyvärinen & Oja, 2000). We formalize this relationship in Proposition 3.3 (proof provided in Appendix B.4).

**Proposition 3.3.** *Let $Z_r$ and $Z_b$ be jointly Gaussian random vectors with marginal covariance matrices $\Sigma_r$ and $\Sigma_b$, and cross-covariance $\Sigma_{rb}$. Define the normalized cross-covariance matrix as $C = \Sigma_r^{-1/2} \Sigma_{rb} \Sigma_b^{-1/2}$. Provided the spectral norm $\|C\|_2$ is sufficiently small, the mutual information admits the following second-order expansion:*

$$I(Z_r; Z_b) = \frac{1}{2} \|C\|_F^2 + o\left(\|C\|_F^2\right), \quad \text{as } \|C\|_2 \to 0.$$

Accordingly, we adopt the cross-covariance penalty as a computationally efficient surrogate for disengtanglement term in Eq. 3. Given centered mini-batch representations $\bar{\mathbf{Z}}_r, \bar{\mathbf{Z}}_b \in \mathbb{R}^{N \times d}$, we compute the empirical cross-covariance matrix $\hat{\Sigma}_{rb} = \frac{1}{N-1} \bar{\mathbf{Z}}_r^\top \bar{\mathbf{Z}}_b$. We then minimize the squared Frobenius norm of $\hat{\Sigma}_{rb}$ to suppress correlations:

$$\mathcal{L}_{\text{disc}} = \|\hat{\Sigma}_{rb}\|_F^2 = \sum_{i=1}^d \sum_{j=1}^d (\hat{\Sigma}_{rb})_{ij}^2. \quad (7)$$

In practice, feature-wise normalization ensures that $\mathbf{Z}_r$ and $\mathbf{Z}_b$ have approximately unit variance along each dimension (Ba et al., 2016). This objective penalizes second-order dependencies, thereby encouraging statistical independence in the learned representations (Zbontar et al., 2021).

**Overall Learning Objective.** We optimize DIBJUDGE end-to-end by minimizing a weighted sum of tractable objectives derived above. Concretely, the final training objective is

$$\mathcal{L} = \mathcal{L}_{\text{task}} + \beta \mathcal{L}_{\text{compress}} + \gamma \mathcal{L}_{\text{bias}} + \lambda \mathcal{L}_{\text{disc}}, \quad (8)$$

where the weights $\beta, \gamma, \lambda$ control the accuracy–compression–bias-capture–independence trade-off.

## 4. Experiments

**Evaluation Benchmarks.** To evaluate the effectiveness of LLM judges across multilingual contexts, we utilize three primary reward modeling benchmarks (Lambert et al., 2024; Son et al., 2024; Gureja et al., 2025) selected to ensure a balanced consideration of the following aspects: a) reasoning and safety alignment across diverse conversational contexts, b) performance across 23 distinct languages, c) the distinction between translated content and native-speaker data. Our primary evaluation metric is accuracy, reported as the category average and the mean of language-specific micro-averages. More details are provided in Appendix C.1.

**Training settings.** We adopt the same training corpus as mR3 (Anugraha et al., 2025) and fine-tune using LoRA (Hu et al., 2022). DIBJUDGE is trained in a single end-to-end stage: all loss terms in Eq. 8 are computed in one forward–backward pass, with $\mathcal{L}_{\text{task}}$ weighted by $1.0$ and the remaining losses by $0.5$. All experiments use Adam (Kingma & Ba, 2015), learning rate $1 \times 10^{-4}$, and maximum sequence length 16384. Further implementation details are provided in Appendix C.2.

**Baselines.** We evaluate DIBJUDGE against proprietary general-purpose models (GPT-4o (Hurst et al., 2024), Gemini-2.5-Flash (Comanici et al., 2025)) and open-source general-purpose LLMs (Qwen2.5/3 (Qwen et al., 2025; Yang et al., 2025)). Since Qwen3 is our backbone, these comparisons isolate gains from our training recipe beyond base model capacity. We additionally benchmark

*Table 1.* Performance evaluation on multilingual reward benchmarks. **Bold** indicates the best performance, and underlined indicates the second-best. Statistical significance compared to the best baseline is denoted by $^\dagger$ ($p < 0.05$) and $^\ddagger$ ($p < 0.01$).

| Model | m-RewardBench (Avg. 23 langs) | RewardBench (English) | MM-Eval (Avg. 18 lang) |
|---|---|---|---|
| *Proprietary Models* | | | |
| GPT-4o (Hurst et al., 2024) | $85.75 \pm 0.42$ | $85.96 \pm 0.35$ | $71.85 \pm 0.81$ |
| Gemini-2.5-Flash (Comanici et al., 2025) | $88.06 \pm 0.49$ | $88.83 \pm 0.47$ | $77.47 \pm 0.76$ |
| *General Open Models* | | | |
| Qwen2.5-3B-Instruct (Qwen et al., 2025) | $66.97 \pm 1.12$ | $68.99 \pm 1.05$ | $57.99 \pm 1.20$ |
| Qwen2.5-7B-Instruct (Qwen et al., 2025) | $77.89 \pm 0.89$ | $78.59 \pm 0.91$ | $65.64 \pm 0.95$ |
| Qwen3-4B (Yang et al., 2025) | $85.06 \pm 0.65$ | $87.54 \pm 0.55$ | $80.85 \pm 0.68$ |
| Qwen3-8B (Yang et al., 2025) | $86.12 \pm 0.52$ | $88.81 \pm 0.48$ | $82.20 \pm 0.60$ |
| *Multilingual Open Reward Models* | | | |
| Nemotron-Multi-49B (Wang et al., 2025b) | $88.83 \pm 0.35$ | $89.71 \pm 0.31$ | $76.31 \pm 0.55$ |
| M-PROMETHEUS 3B (Pombal et al., 2025) | $68.45 \pm 0.98$ | $69.79 \pm 0.92$ | $64.17 \pm 1.10$ |
| M-PROMETHEUS 7B (Pombal et al., 2025) | $78.03 \pm 0.85$ | $76.69 \pm 0.78$ | $69.38 \pm 0.88$ |
| mR3-Qwen3-4B (Anugraha et al., 2025) | $87.21 \pm 0.45$ | $89.75 \pm 0.38$ | $82.55 \pm 0.52$ |
| mR3-Qwen3-8B (Anugraha et al., 2025) | $88.58 \pm 0.41$ | $90.10 \pm 0.40$ | $85.29 \pm 0.45$ |
| Think-as-Locals 7B (Zhang et al., 2026a) | $84.51 \pm 0.60$ | $88.79 \pm 0.52$ | $72.95 \pm 0.70$ |
| *Ours* | | | |
| DIBJudge-Qwen3-4B | $89.84 \pm 0.28^\dagger$ | **$90.32 \pm 0.25$** | $85.16 \pm 0.33$ |
| DIBJudge-Qwen3-8B | **$91.37 \pm 0.22^\ddagger$** | $91.01 \pm 0.20^\dagger$ | **$87.53 \pm 0.28^\ddagger$** |

multilingual reward models/judges, including Nemotron-Multilingual-49B (Wang et al., 2025b), M-Prometheus (3B/7B) (Pombal et al., 2025), mR3 (Anugraha et al., 2025), and Think-as-Locals (7B) (Zhang et al., 2026a).

**Main Results.** Table 1 reports the mean accuracy and standard deviation across benchmarks over three independent runs, with statistical significance assessed using pairwise $t$-tests. On m-RewardBench, DIBJUDGE-Qwen3-8B establishes a new SOTA among open-weight models, significantly outperforming both its backbone-matched counterpart and a substantially larger multilingual baseline. These results confirm the effectiveness of the proposed approach. In terms of generalization, DIBJUDGE-Qwen3-8B achieves superior performance on the English-centric RewardBench, statistically surpassing prior leading methods. This indicates that the proposed method improves multilingual reward modeling without degrading performance on monolingual benchmarks. Detailed results are provided in Appendix D.

## 5. Analysis

We conduct targeted analyses to validate the efficacy, generalization, and internal mechanics of DIBJUDGE. We organize the evaluation around five questions: (i) **RQ1 (Bias Mitigation):** To what extent does DIBJUDGE effectively mitigate translationese bias across languages with varying resource availability? (ii) **RQ2 (Utility Trade-off):** How does the information bottleneck constraint shape the Pareto

Frontier between bias mitigation and downstream task utility? (iii) **RQ3 (Disentanglement):** Do the learned latent representations geometrically disentangle semantic content from translationese artifacts, as theoretically hypothesized? (iv) **RQ4 (Generalization):** Does the model exhibit robustness against unseen bias types (e.g., length bias) that were not explicitly included in the spurious proxy task? (v) **RQ5 (Ablation Study):** How do the distinct components of the DIB objective (Eq. 3) and spurious proxy task contribute to bias mitigation and reward modeling utility? (vi) **RQ6 (Validity Checks):** Can the observed bias and mitigation effects be explained away by reference quality, overfitting, or the pairwise scoring format? Appendix experiments provide additional sensitivity analyses for CAD/SSR (Appx. G.3, G.4), alternative mechanisms (Appx. G.5, G.6), and extended probing (Appx. G.7).

**RQ1: Efficacy in Translationese Bias Mitigation.** We extend the preliminary bias evaluation (§ 2) to a broader suite of domains and datasets. We evaluate performance on three diverse benchmarks: BELEBELE (machine reading comprehension) (Bandarkar et al., 2024), AYA (Singh et al., 2024) (open-ended instruction following), and XL-SUM (Hasan et al., 2021) (summarization). This selection allows us to assess translationese bias across constrained formats and realistic, open-ended interactions. Human-authored references are used as ground-truth targets, while negative samples (rejected responses) are generated via back-translation as described in § 2. To investigate the

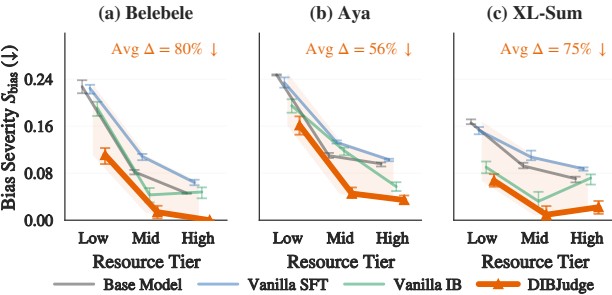

**Figure 4. Bias severity across resource tiers.** $\mathcal{S}_{\text{bias}}$ (lower is better) on BELEBELE, AYA, and XL-SUM. DIBJUDGE reduces bias across all tiers, with average reductions of 80%, 56%, and 75%, and the strongest improvements in Low-Resource settings. Error bars show std over 3 runs; Avg $\Delta$ is relative to *Vanilla SFT*.

impact of data scarcity, we stratify languages into High-, Mid-, and Low-Resource tiers ($n = 10$ languages per tier). We benchmark DIBJUDGE against three baselines: the Base model, Vanilla SFT, and a Vanilla IB variant. We quantify efficacy using the *Bias Severity* metric ($\mathcal{S}_{\text{bias}}$) defined in Equation 1. More details in Appendix. A.2

Figure 4 demonstrates the efficacy of DIBJUDGE in mitigating translationese bias across diverse language resource levels. On the benchmark BELEBELE, DIBJudge achieves a drastic reduction in bias severity, approaching near-zero levels across the mid- and high-resource tiers. This trend extends to generative tasks such as AYA and XL-SUM, where we observe consistent bias suppression. Crucially, DIBJUDGE significantly reduces disparity across resource tiers; whereas vanilla SFT retains marked bias in low-resource settings, our approach effectively dampens these spurious correlations. These findings confirm that DIB-JUDGE targets the bias amplification that disproportionately affects underrepresented languages, rather than merely enhancing general instruction-following capabilities.

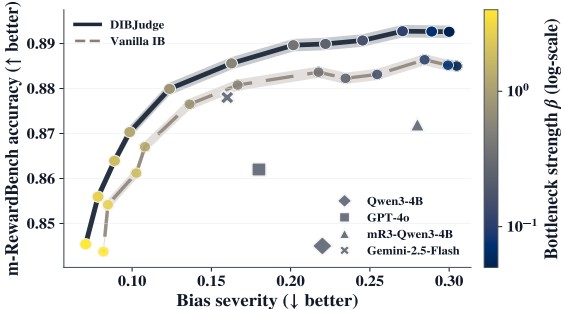

**Figure 5. Bias–utility Pareto Frontier.** Trade-off between Bias Severity ($\downarrow$; x-axis) and m-RewardBench accuracy ($\uparrow$; y-axis). Each point corresponds to a bottleneck strength $\beta$ (log-scaled, color-coded). The resulting Pareto frontiers are traced by DIB-JUDGE (solid) and the VANILLA IB baseline (dashed). DIBJUDGE consistently achieves higher accuracy at comparable bias levels across $\beta$, yielding a uniformly superior bias–utility trade-off. Markers indicate representative SOTA models, which DIBJUDGE outperforms in terms of lower bias and higher accuracy.

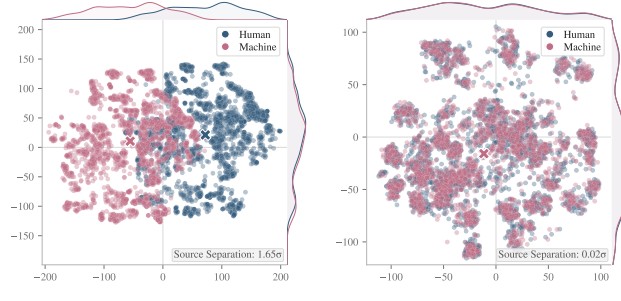

*(a)* Bias Representations ($\mathbf{Z}_b$)    *(b)* Robust Representations ($\mathbf{Z}_r$)

**Figure 6. Visualization of Latent Representation Disentanglement.** t-SNE projections of embeddings for Human (Blue) vs. Machine (Red) text. **(a)** $\mathbf{Z}_b$ clearly separates domains. **(b)** $\mathbf{Z}_r$ shows a mixed distribution, corroborating domain invariance.

**RQ2: The Robustness-Utility Trade-off.** We investigate the tension between robustness and utility by modulating the coefficient $\beta$ of the compression term in Equation 3. Specifically, we aim to characterize the Pareto Frontier of this trade-off. Figure 5 illustrates this dynamic by plotting bias severity ($\mathcal{S}_{\text{bias}}$) against m-RewardBench accuracy. DIBJUDGE achieves a consistently better Pareto frontier than the VANILLA IB baseline, indicating that it learns a more compact and robust representation without discarding key semantic features. Furthermore, DIBJUDGE strictly dominates its base model and strong proprietary baselines (e.g., GPT-4o, Gemini-2.5-Flash), consistently achieving higher accuracy across all fixed levels of bias severity. These findings confirm that the proposed method mitigates translationese bias without substantial utility degradation.

**RQ3: Disentanglement of Latent Representations.** We visualize the geometry of the learned representations using t-SNE (van der Maaten & Hinton, 2008), extracting bias ($\mathbf{Z}_b$) and robust ($\mathbf{Z}_r$) features from a held-out evaluation set comprising human and machine-translated texts. As shown in Figure 6, the latent spaces exhibit divergent topologies. The bias space (Fig. 6a) forms distinct clusters based on text origin, confirming that $\mathbf{Z}_b$ encodes translationese artifacts. Conversely, the robust space (Fig. 6b) demonstrates substantial domain overlap. This phenomenon demonstrates that $\mathbf{Z}_r$ achieves invariance to translationese artifacts, effectively disentangling them from the underlying semantic content.

**RQ4: Zero-Shot Generalization to Unseen Biases.** We evaluate whether DIBJUDGE learns a generally robust reward representation rather than a translationese-only filter. On a held-out subset of *Skywork-Reward-Preference-80K* (Liu et al., 2024a), we test length (Saito et al., 2023) and self-preference (Wataoka et al., 2024) biases unseen during training. As shown in Table 2, DIBJUDGE reduces both the length–score correlation and self-preference bias relative to vanilla SFT/IB. It also improves native-content evaluation: on native multilingual MM-Eval subsets (Linguistics and

*Table 2.* **Zero-Shot Generalization to Unseen Biases.** We evaluate performance on a held-out subset containing biases not encountered during training (Out-of-Distribution). DIBJUDGE achieves the lowest bias scores across both in-distribution (Translationese) and unseen heuristics (Length, Self-Preference).

| | **ID** | **OOD (Unseen Biases)** | |
|---|---|---|---|
| **Method** | *Trans.* $\mathcal{S}_{bias}$ ↓ | *Length* $\rho$ ↓ | *Self-Pref.* $\mathcal{S}_{bias}$ ↓ |
| Vanilla SFT | 0.247 | 0.553 | 0.314 |
| Vanilla IB | 0.168 | 0.482 | 0.276 |
| **DIBJUDGE** | **0.083** | **0.314** | **0.219** |

Language Hallucination), accuracy increases from 76.1% to 82.4%, and on native English RewardBench Chat Hard from 84.47% to 88.10%. A linear probe further confirms that the robust representation removes recoverable origin information (SFT embedding: 82.4%; $\mathbf{Z}_r$: 50.3%, near chance; Appx. G.7).

**RQ5: Impact of DIB Objective Components and Spurious Proxy Tasks.** Ablations confirm that the three DIB objectives are complementary: using compression, bias-capture, and disentanglement together yields the best main-task accuracy (89.85) and the lowest bias score (0.031). The two proxy tasks are also complementary: CLA+LPBC improves bias severity from 0.421 to 0.147 while increasing accuracy from 87.12 to 89.18. We provide the full ablation tables in Appendix G.1 and Appendix G.2.

**RQ6: Quality-Controlled and Practicality Checks.** The preliminary study suggests that multilingual judges often prefer translated responses because they align more closely with English-centric reward-model representations. A possible alternative explanation is that the human references are simply worse than their translated counterparts. We therefore add several controls in Table 3. First, reference-free quality estimates and MQM annotations support the opposite conclusion: human references are generally higher quality, and the cases where automatic differences are not significant do not drive the CAD/SSR patterns. Second, after constructing a quality-matched control subset by mildly degrading human references without introducing translationese, the COMET gap becomes statistically insignificant while the same CAD/SSR effects persist. This separates translationese alignment from generic quality differences.

We also quantify the factors jointly. The mixed-effects regression conditions on the quality gap ΔCOMET while predicting whether a judge selects the machine-translated candidate. CAD and SSR remain significant predictors, whereas ΔCOMET becomes non-significant; moreover, the model using only ΔCOMET has weak explanatory power (AUC .536), while CAD+SSR explains substantially more variance (AUC .692). These results strengthen the causal interpretation used to design the CLA and LPBC proxy tasks: the model should suppress English-anchoring signals

and lexical overlap shortcuts, rather than simply learn a generic preference for fluent text.

Finally, we test whether the method remains useful outside the exact translationese benchmark construction. DIBJUDGE improves native multilingual evaluation and native English Chat Hard accuracy, retains most performance with 40% of the training data, reproduces under independent retraining, and remains competitive in a pointwise scoring format. We define the English centroid using human-authored English references rather than machine English; an EN→ZH→EN stress test confirms that machine English is still degraded (sacreBLEU 40.09, BLEURT 0.6445). For the bias metric, a fully unbiased judge has $\mathcal{S}_{bias}=0$ because severity counts consistent machine wins among order-consistent decisions, while 0.5 would indicate random preference inside that conditioned subset.

The robustness improvements are not confined to native-only cases. On translation-based MM-Eval Chat/Reasoning/Safety subsets, DIBJUDGE-Qwen3-8B improves from 90.7% to 92.5%, showing that debiasing does not sacrifice the translated evaluation settings used by current multilingual reward models. The 40% data experiment further suggests that the proxy supervision is sample efficient: the 40k-instance model remains within 1.81 points on m-RewardBench and 1.57 points on MM-Eval of the full-data model. Independent retraining gives 89.68±0.30 / 90.71±0.17 / 84.91±0.36 on m-RewardBench / Reward-Bench / MM-Eval, matching the original 89.84±0.28 / 90.32±0.25 / 85.16±0.33 within variance. Finally, the pointwise variant (89.21±0.41 / 90.46±0.36 / 84.48±0.29) remains close to the pairwise model, indicating that the learned robust representation is not tied to response-pair comparison alone.

These controls also clarify what is, and is not, required by our training signal. The CLA objective uses the target-language-to-English translated-English manifold to expose English anchoring, but the spurious variable does not require gold parallel data: translated variants are generated automatically and are used only to construct auxiliary origin and alignment signals. LPBC complements CLA by targeting lexical-overlap shortcuts inside each language, which explains why the combined proxy improves both bias severity and reward accuracy relative to either component alone. In representation space, this design yields the intended separation: origin information remains highly recoverable from $\mathbf{Z}_b$ (96.1% probe accuracy in Appendix G.7) but is removed from $\mathbf{Z}_r$ (50.3%, near chance). Thus, reduced $\mathcal{S}_{bias}$ reflects fewer systematic machine wins under controlled semantic comparisons rather than indiscriminate preference against machine translations. This distinction is most important for low-resource languages, where scarce native data amplify English anchoring and surface-overlap

*Table 3.* **Additional validity checks integrated into the main analysis.** The diagnostics rule out quality confounds, test robustness beyond translated benchmarks, and verify that DIBJUDGE's gains are not an artifact of a particular data scale or pairwise objective.

| Check | Setup | Main finding |
|---|---|---|
| Reference quality | COMET-Kiwi over all 29 languages; MQM on 200 Chinese–English cases. | Human references are higher quality in 23/29 languages ($p<.05$; other six n.s.); MQM favors human text by 2.83 points (human 2.58 vs. machine 5.41, 95% CI [2.28, 3.39], $p=10^{-4}$; 188/200 human wins). |
| Quality-matched attribution | Six-language subset (zh/ja/ar/ko/th/vi; 100 cases each) with mild non-translationese degradation of human references. | After matching quality (e.g., Chinese $\Delta$COMET 2.69, $p<.001 \rightarrow 0.48$, $p=.218$), CAD/SSR trends remain, indicating that the discovered factors are not merely artifacts of lower machine-translation quality. |
| Regression control | Mixed-effects logistic model predicting machine wins from CAD, SSR, and $\Delta$COMET. | CAD ($\beta=-0.57$, SE .13) and SSR ($\beta=-0.79$, SE .15) remain significant ($p<.001$), while $\Delta$COMET is not ($p=.520$); AUC rises from .536 ($\Delta$COMET only) to .692 (CAD+SSR) and .701 (full). |
| Native generalization | Native MM-Eval subsets and native English RewardBench Chat Hard. | DIBJUDGE-Qwen3-8B improves native multilingual Linguistics/Language-Hallucination accuracy from 76.1% to 82.4%, and native English Chat Hard from 84.47% to 88.10%. |
| Efficiency/stability | 40% training data, independent retraining, and pointwise scoring. | With only 40k instances, performance remains close to full data (m-RewardBench 88.03 vs. 89.84; MM-Eval 83.59 vs. 85.16). Retraining reproduces the original scores within standard deviations; pointwise scoring remains close to pairwise scoring (89.21/90.46/84.48 vs. 89.84/90.32/85.16). |

shortcuts; maintaining native and translated benchmark accuracy while reducing those shortcuts indicates that the method suppresses a harmful evaluation artifact rather than cross-lingual transfer itself.

# 6. Related Work

**LLM-as-a-Judge.** The *LLM-as-a-Judge* paradigm marks a fundamental shift from traditional $n$-gram (e.g., BLEU (Papineni et al., 2002), ROUGE (Lin, 2004)) and embedding-based metrics (e.g., BERTScore (Zhang et al., 2020b; Rei et al., 2020)) toward generative evaluation (Gu et al., 2024; Li et al., 2025). While early adoption relied on proprietary models like GPT-4 due to their high correlation with human judgment (Liu et al., 2023; Zheng et al., 2023), concerns regarding cost and transparency have catalyzed a transition to open-weight evaluators (Wang et al., 2024e;c;a; Kim et al., 2024a;b). Recently, this paradigm has further evolved from direct generative to incorporating explicit reasoning steps to enhance reliability (Chen et al., 2025a; Guo et al., 2025; Chen et al., 2025b; Zhang et al., 2026b). However, despite these advancements, LLM judges remain susceptible to systematic biases (Ye et al., 2025; Wang et al., 2024b; Zheng et al., 2024), such as position bias (Shi et al., 2025; Ko et al., 2020; Li et al., 2024), verbosity bias (Saito et al., 2023), and self- preference bias (Wang et al., 2024c). In contrast to prior work that primarily studies bias in English-centric settings, we investigate *translationese bias* in multilingual contexts and analyze the spurious correlations underlying it.

**Multilingual Judges.** Compared to the English context, multilingual LLM-as-a-Judge and culture-sensitive multilingual evaluation remain significantly underexplored (Wu et al., 2026). Initial efforts to bridge this gap, such as Hercule (Doddapaneni et al., 2025) and M-Prometheus (Pombal et al., 2025), rely heavily on fine-tuning with translated or synthetic instruction sets. More recently, approaches like mR3 (Anugraha et al., 2025), Think-as-Locals (Zhang et al., 2026a), and UniRRM (Lai et al., 2026) have advanced the field by integrating reasoning capabilities, employing Chain-of-Thought (CoT) (Wei et al., 2022) distillation and reinforcement learning to enhance multilingual reward modeling. However, despite these achieving promising results, the robustness of these evaluators remains unexamined. Crucially, existing frameworks fail to account for the systematic artifacts introduced by translation-based training data. To address this reliability gap, our work provides the first dedicated mitigation of *translationese bias*, resolving specific failures in cross-lingual evaluation that prior methodologies overlook.

**Information Bottleneck in LLMs.** Originally formulated to extract minimal sufficient statistics (Tishby et al., 2000), the Information Bottleneck (IB) principle has recently been used to analyze and optimize LLMs: mapping hidden states to human-readable concepts (Sun et al., 2025; Li et al., 2023), making CoT paths invariant to prompt nuances (Lei et al., 2025), filtering noisy Retrieval-Augmented Generation contexts (Zhu et al., 2024), and stripping adversarial triggers via IBProtector (Liu et al., 2024b). In contrast, we apply a disentangled IB to debias LLM judges.

# 7. Conclusion

We studied translationese bias in multilingual LLM-as-a-Judge systems, where English-aligned machine translations can be favored over human text, especially in low-resource languages. We introduced DIBJUDGE, a disentangled information-bottleneck framework that separates judgment-relevant semantics from translationese shortcuts. Across multilingual reward-modeling and bias benchmarks, DIBJUDGE reduces this bias while preserving strong utility. Validity checks confirm that these gains persist under quality controls and native-language evaluation.

## Acknowledgment

This work was supported in part by the Science Fund for Creative Research Groups of the National Natural Science Foundation of China under Grant 62521006, in part by the National Natural Science Foundation of China (62276077, 62406091, U23B2055, 62350710797), in part by Guangdong S&T Program (2024B0101050003), in part by the Guangdong Basic and Applied Basic Research Foundation (2024A1515011205, 2026A1515011718), and in part by Shenzhen Science and Technology Program (KQTD20240729102154066).

## Impact Statement

This paper presents work whose goal is to advance the field of machine learning, with a focus on improving the robustness of multilingual evaluation using large language models. The methods proposed in this work are intended for model evaluation and benchmarking rather than direct user-facing applications. While improved evaluation may have downstream benefits for the development of more reliable and inclusive multilingual systems, we do not foresee significant or immediate negative societal impacts arising from this work.

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

# Appendix Contents

# A. Bias Evaluation Suite

## A.1. Language Selection and Taxonomy

We adopt the language taxonomy proposed by Joshi et al. (2020) to categorize languages based on their resource availability. Specifically, we partition the selected languages into three distinct tiers based on their assigned resource classes: *High-resource* (Classes 4 and 5), *Mid-resource* (Class 3), and *Low-resource* (Classes 0, 1, and 2).

Our evaluation spans three primary datasets: Aya (Singh et al., 2024), Belebele (Bandarkar et al., 2024), and XL-Sum (Hasan et al., 2021). The complete distribution of evaluated languages across these resource tiers is summarized in Table 4.

*Table 4.* Classification of evaluated languages across datasets based on Joshi et al. (2020) taxonomy.

| Dataset | High-Resource | Mid-Resource | Low-Resource |
| --- | --- | --- | --- |
| **Aya** | Basque, English, Finnish, Hindi, Japanese, Portuguese, Simp. Chinese, Spanish, Arabic, Vietnamese | Bengali, Cebuano, Filipino, Indonesian, Lithuanian, Malay, Tamil, Thai, Ukrainian, Urdu | Amharic, Irish, Kyrgyz, Nepali, Malagasy, Sinhala, S. Pashto, Telugu, Yoruba, Zulu |
| **Belebele** | Arabic, English, Finnish, Hindi, Japanese, Korean, Russian, Turkish, Vietnamese, Simp. Chinese | Bengali, Greek, Hebrew, Georgian, Kazakh, Tamil, Thai, Ukrainian, Urdu, Malay | Amharic, Tibetan, Guarani, Kannada, Khmer, Kyrgyz, Burmese, Punjabi, Pashto, Zulu |
| **XL-Sum** | Arabic, Simp. Chinese, English, French, Hindi, Japanese, Korean, Russian, Turkish, Vietnamese | Azerbaijani, Bengali, Indonesian, Tamil, Thai, Ukrainian, Urdu, Uzbek | Amharic, Burmese, Hausa, Kyrgyz, Marathi, Nepali, Pashto, Sinhala, Telugu, Welsh |

## A.2. Test Set Construction

We formulate a pairwise preference task where an LLM evaluator compares two candidate responses for a given query: (i) **Chosen** ($x_H$): The original human-authored or high-quality translated reference; and (ii) **Rejected** ($x_M$): A machine-generated counterpart produced via standard back-translation ($L_{target} \rightarrow$ English$\rightarrow L_{target}$) using NLLB-200-3.3B (Costa-Jussà et al., 2022)[2] to inject subtle translationese artifacts.

To isolate translationese as the primary variable and mitigate length-based confounding, we enforce a length constraint where the token count differential between $x_H$ and $x_M$ is within $\pm 5\%$. We further ensure evaluation robustness through a position-swapping protocol, retaining only *consistent judgments* where the model's preference remains invariant to the presentation order.

---

[2]https://huggingface.co/facebook/nllb-200-3.3B

**Consistency and bias severity.**    The consistency filter separates systematic preference from arbitrary or position-driven flips. Bias severity is therefore interpreted as the rate at which the judge consistently favors translationese over the human reference; an unbiased judge has $\mathcal{S}_{\text{bias}} = 0$, not $0.5$, because the human reference is the higher-quality target in this controlled benchmark. Empirically, inconsistency and bias severity capture distinct phenomena: languages with similar inconsistency ratios can exhibit very different $\mathcal{S}_{\text{bias}}$, so the consistency filter does not collapse low-resource uncertainty into the bias metric.

Beyond the baseline comparison, we introduce two distinct experimental configurations (summarized with examples in the following subsection A.3):

- **Parallel**: Both candidates are semantically equivalent and factually correct. This isolates the model's stylistic preference for human vs. machine-translated syntax.

- **Perturbed**: We introduce minor, controlled edits to the machine-generated candidate ($x_M$) to alter its factual correctness or logical coherence, testing whether the judge prioritizes correctness over stylistic fluency.

## A.3. Detailed Dataset Examples

In this section, we provide the full text for the construction of our test sets across the three primary datasets: Aya, Belebele, and XL-Sum. For each dataset, we present both the **Parallel** setting (testing stylistic preference/bias) and the **Perturbed** setting (testing sensitivity to factual correctness).

### Example: Aya Dataset

---

**User Question:**
问题：油的营养价值如何？

---

**Setting 1: Parallel (Stylistic Variation)**
*Both answers are factually correct; B uses back-translated "translationese."*

**[Assistant A (Chosen $x_H$)]**
回答：油是脂肪的重要来源，提供能量和必需脂肪酸，但摄入应适量控制，避免摄入过多的脂肪。

**[Assistant B (Rejected $x_M$)]**
回答：油脂乃脂肪主来源，供应热能与必备脂肪酸，然食量宜适当限制，防止摄取过多的脂质。

---

**Setting 2: Perturbed (Factual Error)**
*Assistant B is modified to suggest an incorrect health outcome.*

**[Assistant A (Chosen $x_H$)]**
回答：油是脂肪的重要来源，提供能量和必需脂肪酸，但摄入应适量控制，避免摄入过多的脂肪。

**[Assistant B (Rejected $x_M$)]**
回答：油是脂肪的重要来源，提供能量和必需脂肪酸，但摄入应适量控制，以促进脂肪摄入。

---

### Example: Belebele Dataset

---

**User Question (System Prompt):**
Create an example of multiple-choice reading test in Chinese (Simplified). You may choose any topic you want...
Output only the test (passage, questions, answer choices, and correct answer).

---

---

**Setting 1: Parallel**
**[Assistant A (Chosen $x_H$)]**
### Passage: 乘坐头等舱或商务舱最明显的方式是... 航空公司非常清楚，有一些核心乘客愿花高价...
### Question: 根据这段文字，预订某些票价舱位时，搜索什么是浪费时间？
### Answer Choices: (1) 直达航班(2) 商务舱(3) 折扣(4) 头等舱
### Correct Answer: (3) 折扣

**[Assistant B (Rejected $x_M$)]**
### Passage: 想体验头等舱或商务舱最直接的途径... 航司深知，部分核心客户乐意支付高昂费用...
### Question: 依照本文内容，在订特定票价舱位时，查找何事乃浪费光阴？
### Answer Choices: (1) 不停航程(2) 公务舱(3) 优惠(4) 首等舱
### Correct Answer: (3) 优惠

---

**Setting 2: Perturbed**
*Assistant B's "Correct Answer" index is modified to be factually wrong.*
**[Assistant B (Rejected $x_M$)]**
... ### Answer Choices: (1) 不停航程(2) 公务舱(3) 优惠(4) 首等舱
### Correct Answer: (4) 首等舱

**Example: XL-Sum Dataset**

---

**User Question (Summarization Task):**
Generate a concise, coherent abstractive summary in Chinese_Simplified... Do not include information not present in the source text.
*Source Text: [Long article regarding the Brazilian "Operation Weak Flesh" meat scandal involving President Michel Temer, JBS, and BRF...]*

---

**Setting 1: Parallel (Stylistic Variation)**
*Both summaries accurately reflect that countries suspended imports for several years.*

**[Assistant A (Chosen $x_H$)]**
近期巴西"问题肉"丑闻揭露巴西的一些公司已经数年销售不安全肉类产品。包括中国在内的一些国家和组织已经叫停巴西的进口肉。

**[Assistant B (Rejected $x_M$)]**
最近巴西"劣质肉"风波披露，部分巴西公司长达数年贩卖问题肉品。中国等许多国家与机构，均已暂停来自巴西的肉类进口。

---

**Setting 2: Perturbed (Factual Hallucination)**
*Assistant B is modified to falsely state that restrictions were "relaxed" (放宽限制) and that the scandal "just started" (刚开始).*

**[Assistant A (Chosen $x_H$)]**
近期巴西"问题肉"丑闻揭露巴西的一些公司已经数年销售不安全肉类产品。包括中国在内的一些国家和组织已经叫停巴西的进口肉。

[Assistant B (Rejected $x_M$)]
近期巴西"问题肉"丑闻揭露巴西的一些公司刚开始销售不安全肉类产品。包括中国在内的一些国家和组织已经放宽限制巴西的进口肉。

## B. Theory Supplementary

### B.1. Proof of The Upper Boundary of $I(X; Z_r)$

*Proof.* By definition, the mutual information $I(X; Z_r)$ is the KL divergence between the joint distribution $q_\phi(X, Z_r) = p(x)q_\phi(z_r|x)$ and the product of the marginals $p(x)q_\phi(z_r)$, where $q_\phi(z_r) = \int q_\phi(z_r|x)p(x)\,dx$ is the aggregate posterior (marginal) distribution. This is expressed as:

$$I(X; Z_r) = \mathbb{E}_{x \sim p(x)} \left[ \int q_\phi(z_r|x) \log \frac{q_\phi(z_r|x)}{q_\phi(z_r)}\,dz_r \right]. \tag{9}$$

To derive the upper bound, we introduce an arbitrary fixed prior $p(z_r)$. We multiply and divide the argument of the logarithm by this prior $p(z_r)$:

$$\begin{aligned}
I(X; Z_r) &= \mathbb{E}_{x \sim p(x)} \left[ \int q_\phi(z_r|x) \log \left( \frac{q_\phi(z_r|x)}{q_\phi(z_r)} \cdot \frac{p(z_r)}{p(z_r)} \right) dz_r \right] \\
&= \mathbb{E}_{x \sim p(x)} \left[ \int q_\phi(z_r|x) \left( \log \frac{q_\phi(z_r|x)}{p(z_r)} - \log \frac{q_\phi(z_r)}{p(z_r)} \right) dz_r \right].
\end{aligned} \tag{10}$$

Using the linearity of the expectation, we separate the integral into two terms:

$$I(X; Z_r) = \mathbb{E}_{x \sim p(x)} \left[ \int q_\phi(z_r|x) \log \frac{q_\phi(z_r|x)}{p(z_r)}\,dz_r \right] - \mathbb{E}_{x \sim p(x)} \left[ \int q_\phi(z_r|x) \log \frac{q_\phi(z_r)}{p(z_r)}\,dz_r \right]. \tag{11}$$

The first term is exactly the expected KL divergence between the posterior and the prior. For the second term, we observe that $\log \frac{q_\phi(z_r)}{p(z_r)}$ does not depend on $x$ directly, other than through the integration of the joint density. We can simplify the expectation over $x$:

$$\begin{aligned}
\mathbb{E}_{x \sim p(x)} \left[ \int q_\phi(z_r|x) \log \frac{q_\phi(z_r)}{p(z_r)}\,dz_r \right] &= \int \left( \int p(x)q_\phi(z_r|x)\,dx \right) \log \frac{q_\phi(z_r)}{p(z_r)}\,dz_r \\
&= \int q_\phi(z_r) \log \frac{q_\phi(z_r)}{p(z_r)}\,dz_r \\
&= D_{\mathrm{KL}}(q_\phi(Z_r) \| p(Z_r)).
\end{aligned} \tag{12}$$

Substituting this back into the expression for mutual information yields the following decomposition:

$$I(X; Z_r) = \mathbb{E}_{x \sim p(x)} \left[ D_{\mathrm{KL}}(q_\phi(Z_r|x) \| p(Z_r)) \right] - D_{\mathrm{KL}}(q_\phi(Z_r) \| p(Z_r)). \tag{13}$$

Since the KL divergence is non-negative (Gibbs' inequality), i.e., $D_{\mathrm{KL}}(q_\phi(Z_r) \| p(Z_r)) \geq 0$, it follows that:

$$I(X; Z_r) \leq \mathbb{E}_{x \sim p(x)} \left[ D_{\mathrm{KL}}(q_\phi(Z_r|x) \| p(Z_r)) \right]. \tag{14}$$

This completes the proof. $\qquad \square$

### B.2. Detailed Derivation of $\mathcal{L}_{\text{compress}}$

We derive the analytic form of the compression regularizer used in Eq. (4). At each time step $t \in \{1, \ldots, T\}$, we assume a diagonal-covariance Gaussian variational posterior

$$q_\phi(\mathbf{z}_{r,t} \mid x) = \mathcal{N}(\boldsymbol{\mu}_t, \mathrm{diag}(\boldsymbol{\sigma}_t^2)), \tag{15}$$

and a standard Gaussian prior

$$p(\mathbf{z}_{r,t}) = \mathcal{N}(\mathbf{0}, \mathbf{I}), \tag{16}$$

where $\boldsymbol{\mu}_t \in \mathbb{R}^d$ and $\boldsymbol{\sigma}_t^2 \in \mathbb{R}_{>0}^d$.

We regularize the information capacity by minimizing the average KL divergence across the sequence:

$$\mathcal{L}_{\text{compress}} = \frac{1}{T} \sum_{t=1}^{T} D_{\text{KL}}(q_\phi(\mathbf{z}_{r,t} \mid x) \,\|\, p(\mathbf{z}_{r,t})). \tag{17}$$

Thus, it suffices to derive a closed form for $D_{\text{KL}}\big(\mathcal{N}(\boldsymbol{\mu}_t, \text{diag}(\boldsymbol{\sigma}_t^2)) \,\|\, \mathcal{N}(\mathbf{0}, \mathbf{I})\big)$.

For $q = \mathcal{N}(\boldsymbol{\mu}_q, \boldsymbol{\Sigma}_q)$ and $p = \mathcal{N}(\boldsymbol{\mu}_p, \boldsymbol{\Sigma}_p)$ in $\mathbb{R}^d$, the KL divergence admits the well-known closed form:

$$D_{\text{KL}}(q\|p) = \frac{1}{2}\left(\log\frac{|\boldsymbol{\Sigma}_p|}{|\boldsymbol{\Sigma}_q|} - d + \text{tr}\big(\boldsymbol{\Sigma}_p^{-1}\boldsymbol{\Sigma}_q\big) + (\boldsymbol{\mu}_p - \boldsymbol{\mu}_q)^\top \boldsymbol{\Sigma}_p^{-1}(\boldsymbol{\mu}_p - \boldsymbol{\mu}_q)\right). \tag{18}$$

In our case, $\boldsymbol{\mu}_q = \boldsymbol{\mu}_t$, $\boldsymbol{\Sigma}_q = \text{diag}(\boldsymbol{\sigma}_t^2)$, $\boldsymbol{\mu}_p = \mathbf{0}$, and $\boldsymbol{\Sigma}_p = \mathbf{I}$.

Since $\boldsymbol{\Sigma}_p = \mathbf{I}$, we have $|\boldsymbol{\Sigma}_p| = 1$ and $\boldsymbol{\Sigma}_p^{-1} = \mathbf{I}$. Plugging into Eq. (18) yields

$$D_{\text{KL}}(q\|p) = \frac{1}{2}\left(\log\frac{1}{|\boldsymbol{\Sigma}_q|} - d + \text{tr}(\boldsymbol{\Sigma}_q) + \boldsymbol{\mu}_t^\top \boldsymbol{\mu}_t\right). \tag{19}$$

Because $\boldsymbol{\Sigma}_q = \text{diag}(\boldsymbol{\sigma}_t^2)$ is diagonal,

$$|\boldsymbol{\Sigma}_q| = \prod_{j=1}^{d} \sigma_{t,j}^2, \tag{20}$$

$$\log|\boldsymbol{\Sigma}_q| = \sum_{j=1}^{d} \log \sigma_{t,j}^2, \tag{21}$$

$$\text{tr}(\boldsymbol{\Sigma}_q) = \sum_{j=1}^{d} \sigma_{t,j}^2, \tag{22}$$

$$\boldsymbol{\mu}_t^\top \boldsymbol{\mu}_t = \sum_{j=1}^{d} \mu_{t,j}^2. \tag{23}$$

Substituting these into Eq. (19), we obtain

$$D_{\text{KL}}\big(\mathcal{N}(\boldsymbol{\mu}_t, \text{diag}(\boldsymbol{\sigma}_t^2)) \,\|\, \mathcal{N}(\mathbf{0}, \mathbf{I})\big) = \frac{1}{2}\sum_{j=1}^{d}\left(\sigma_{t,j}^2 + \mu_{t,j}^2 - 1 - \log \sigma_{t,j}^2\right). \tag{24}$$

Rearranging Eq. (24) gives the equivalent expression

$$D_{\text{KL}} = -\frac{1}{2}\sum_{j=1}^{d}\left(1 + \log \sigma_{t,j}^2 - \mu_{t,j}^2 - \sigma_{t,j}^2\right). \tag{25}$$

Finally, averaging the KL divergence across the sequence as defined in Eq. (17), we obtain

$$\mathcal{L}_{\text{compress}} = \frac{1}{T}\sum_{t=1}^{T} D_{\text{KL}}\big(\mathcal{N}(\boldsymbol{\mu}_t, \text{diag}(\boldsymbol{\sigma}_t^2)) \,\|\, \mathcal{N}(\mathbf{0}, \mathbf{I})\big) \tag{26}$$

$$= \frac{1}{2T}\sum_{t=1}^{T}\sum_{j=1}^{d}\left(\sigma_{t,j}^2 + \mu_{t,j}^2 - 1 - \log \sigma_{t,j}^2\right) \tag{27}$$

$$= -\frac{1}{2T}\sum_{t=1}^{T}\sum_{j=1}^{d}\left(1 + \log \sigma_{t,j}^2 - \mu_{t,j}^2 - \sigma_{t,j}^2\right), \tag{28}$$

which matches Eq. (4).

## B.3. Proof of the Variational Lower Bound on $I(U;V)$

*Proof.* By definition, the mutual information $I(U;V)$ can be expressed as the difference between the marginal entropy of $U$ and the conditional entropy of $U$ given $V$:

$$I(U;V) = H(U) - H(U|V). \tag{29}$$

The conditional entropy is defined as the expectation of the negative log-probability of the true conditional distribution $p(u|v)$:

$$H(U|V) = \mathbb{E}_{u,v \sim p(u,v)}[-\log p(u|v)]. \tag{30}$$

Substituting this back into the expression for mutual information yields:

$$I(U;V) = H(U) + \mathbb{E}_{u,v \sim p(u,v)}[\log p(u|v)]. \tag{31}$$

Since the true conditional distribution $p(u|v)$ is often unknown or intractable, we introduce a variational approximation $q_\theta(u|v)$. We consider the expected Kullback-Leibler (KL) divergence between the true conditional distribution and the variational approximation:

$$\mathbb{E}_{v \sim p(v)}\left[D_{\mathrm{KL}}(p(U|v)\|q_\theta(U|v))\right] = \mathbb{E}_{u,v \sim p(u,v)}\left[\log \frac{p(u|v)}{q_\theta(u|v)}\right]. \tag{32}$$

By the non-negativity of the KL divergence, we have:

$$\mathbb{E}_{u,v \sim p(u,v)}[\log p(u|v) - \log q_\theta(u|v)] \geq 0, \tag{33}$$

which implies:

$$\mathbb{E}_{u,v \sim p(u,v)}[\log p(u|v)] \geq \mathbb{E}_{u,v \sim p(u,v)}[\log q_\theta(u|v)]. \tag{34}$$

Finally, substituting the inequality (34) into (31), we obtain the lower bound:

$$\begin{aligned} I(U;V) &= H(U) + \mathbb{E}_{u,v \sim p(u,v)}[\log p(u|v)] \\ &\geq H(U) + \mathbb{E}_{u,v \sim p(u,v)}[\log q_\theta(u|v)]. \end{aligned} \tag{35}$$

Consequently, maximizing the expected log-likelihood of the variational distribution $q_\theta(u|v)$ maximizes the lower bound of the mutual information $I(U;V)$. $\qquad\square$

## B.4. Relationship Between Mutual Information and Cross-Covariance

*Proof.* Let $(Z_r, Z_b)$ be jointly Gaussian with mean 0 (without loss of generality, since mutual information is invariant under translations) and block covariance

$$\Sigma = \begin{pmatrix} \Sigma_r & \Sigma_{rb} \\ \Sigma_{br} & \Sigma_b \end{pmatrix}, \qquad \Sigma_{br} = \Sigma_{rb}^\top,$$

where $\Sigma_r \succ 0$ and $\Sigma_b \succ 0$ so that $\Sigma_r^{-1/2}$ and $\Sigma_b^{-1/2}$ are well-defined. Define

$$C := \Sigma_r^{-1/2}\Sigma_{rb}\Sigma_b^{-1/2}.$$

**Step 1: Mutual information for Gaussians from the definition.** By definition,

$$I(Z_r; Z_b) = \mathbb{E}\left[\log \frac{p_{Z_r,Z_b}(Z_r, Z_b)}{p_{Z_r}(Z_r)\, p_{Z_b}(Z_b)}\right].$$

For a centered $d$-dimensional Gaussian $X \sim \mathcal{N}(0, \Sigma_X)$ with $\Sigma_X \succ 0$, its density is

$$p_X(x) = (2\pi)^{-d/2}(\det \Sigma_X)^{-1/2} \exp\left(-\tfrac{1}{2}x^\top \Sigma_X^{-1} x\right).$$

Applying this to $(Z_r, Z_b)$ and to the marginals $Z_r$ and $Z_b$, we obtain

$$\log \frac{p_{Z_r, Z_b}(z_r, z_b)}{p_{Z_r}(z_r) p_{Z_b}(z_b)} = -\frac{1}{2} \log \det \Sigma + \frac{1}{2} \log \det \Sigma_r + \frac{1}{2} \log \det \Sigma_b$$
$$-\frac{1}{2} \begin{pmatrix} z_r \\ z_b \end{pmatrix}^\top \Sigma^{-1} \begin{pmatrix} z_r \\ z_b \end{pmatrix} + \frac{1}{2} z_r^\top \Sigma_r^{-1} z_r + \frac{1}{2} z_b^\top \Sigma_b^{-1} z_b.$$

Taking expectation under the joint law of $(Z_r, Z_b)$ yields

$$I(Z_r; Z_b) = -\frac{1}{2} \log \det \Sigma + \frac{1}{2} \log \det \Sigma_r + \frac{1}{2} \log \det \Sigma_b$$
$$-\frac{1}{2} \mathbb{E}\left[\begin{pmatrix} Z_r \\ Z_b \end{pmatrix}^\top \Sigma^{-1} \begin{pmatrix} Z_r \\ Z_b \end{pmatrix}\right] + \frac{1}{2} \mathbb{E}[Z_r^\top \Sigma_r^{-1} Z_r] + \frac{1}{2} \mathbb{E}[Z_b^\top \Sigma_b^{-1} Z_b].$$

Using the identity $\mathbb{E}[X^\top A X] = \operatorname{tr}(A \operatorname{Cov}(X))$ for any centered random vector $X$ with finite second moment and any matrix $A$ of compatible size, we get

$$\mathbb{E}\left[\begin{pmatrix} Z_r \\ Z_b \end{pmatrix}^\top \Sigma^{-1} \begin{pmatrix} Z_r \\ Z_b \end{pmatrix}\right] = \operatorname{tr}(\Sigma^{-1}\Sigma) = \operatorname{tr}(I) = d_r + d_b,$$

$$\mathbb{E}[Z_r^\top \Sigma_r^{-1} Z_r] = \operatorname{tr}(\Sigma_r^{-1}\Sigma_r) = d_r, \qquad \mathbb{E}[Z_b^\top \Sigma_b^{-1} Z_b] = \operatorname{tr}(\Sigma_b^{-1}\Sigma_b) = d_b,$$

so the quadratic terms cancel. Hence

$$I(Z_r; Z_b) = \frac{1}{2} \log \frac{\det \Sigma_r \; \det \Sigma_b}{\det \Sigma}. \tag{36}$$

**Step 2: Expressing $\det \Sigma$ in terms of $C$.** By the block determinant (Schur complement) formula with $\Sigma_b \succ 0$,

$$\det \Sigma = \det(\Sigma_b) \, \det(\Sigma_r - \Sigma_{rb} \Sigma_b^{-1} \Sigma_{br}).$$

Next,

$$\Sigma_r - \Sigma_{rb} \Sigma_b^{-1} \Sigma_{br} = \Sigma_r^{1/2} \Big( I - \Sigma_r^{-1/2} \Sigma_{rb} \Sigma_b^{-1} \Sigma_{br} \Sigma_r^{-1/2} \Big) \Sigma_r^{1/2}$$
$$= \Sigma_r^{1/2} \Big( I - \Sigma_r^{-1/2} \Sigma_{rb} \Sigma_b^{-1/2} \, \Sigma_b^{-1/2} \Sigma_{br} \Sigma_r^{-1/2} \Big) \Sigma_r^{1/2}$$
$$= \Sigma_r^{1/2} \big( I - C C^\top \big) \Sigma_r^{1/2}.$$

Therefore,

$$\det(\Sigma_r - \Sigma_{rb} \Sigma_b^{-1} \Sigma_{br}) = \det(\Sigma_r) \, \det(I - C C^\top),$$

and thus

$$\det \Sigma = \det(\Sigma_r) \det(\Sigma_b) \det(I - C C^\top).$$

Substituting into (36) gives

$$I(Z_r; Z_b) = -\frac{1}{2} \log \det(I - C C^\top). \tag{37}$$

**Step 3: Second-order expansion for small $\|C\|_2$.** Let $A := C C^\top$. Then $A \succeq 0$ and $\|A\|_2 = \|C C^\top\|_2 = \|C\|_2^2$. Assume $\|C\|_2$ is sufficiently small so that $\|A\|_2 < 1$. In this regime, the matrix power series for the principal logarithm holds:

$$\log(I - A) = -\sum_{k=1}^{\infty} \frac{A^k}{k}, \qquad \text{(convergent in operator norm since } \|A\|_2 < 1).$$

Taking traces and using (37) together with $\log \det(I - A) = \operatorname{tr}(\log(I - A))$ yields

$$I(Z_r; Z_b) = \frac{1}{2} \sum_{k=1}^{\infty} \frac{\operatorname{tr}(A^k)}{k}. \tag{38}$$

The leading term is

$$\frac{1}{2}\operatorname{tr}(A) = \frac{1}{2}\operatorname{tr}(CC^\top) = \frac{1}{2}\|C\|_F^2.$$

It remains to show that the remainder is $o(\|C\|_F^2)$ as $\|C\|_2 \to 0$. Since $A \succeq 0$ has eigenvalues $\{\lambda_i\}_{i=1}^m$ (with $m = \operatorname{rank}(A)$) in $[0, \|A\|_2]$, we have for every $k \geq 2$,

$$\operatorname{tr}(A^k) = \sum_{i=1}^m \lambda_i^k \leq \left(\max_i \lambda_i\right)^{k-1} \sum_{i=1}^m \lambda_i = \|A\|_2^{k-1} \operatorname{tr}(A).$$

Therefore, the tail of (38) satisfies

$$0 \leq \sum_{k=2}^\infty \frac{\operatorname{tr}(A^k)}{k} \leq \operatorname{tr}(A) \sum_{k=2}^\infty \frac{\|A\|_2^{k-1}}{k} \leq \operatorname{tr}(A) \sum_{k=2}^\infty \|A\|_2^{k-1} = \operatorname{tr}(A) \frac{\|A\|_2}{1 - \|A\|_2}.$$

Since $\|A\|_2 = \|C\|_2^2 \to 0$, we have $\frac{\|A\|_2}{1-\|A\|_2} \to 0$, and hence

$$\sum_{k=2}^\infty \frac{\operatorname{tr}(A^k)}{k} = o\big(\operatorname{tr}(A)\big) = o\big(\|C\|_F^2\big) \quad \text{as } \|C\|_2 \to 0.$$

Combining this with (38) yields

$$I(Z_r; Z_b) = \frac{1}{2}\|C\|_F^2 + o\big(\|C\|_F^2\big), \qquad \text{as } \|C\|_2 \to 0,$$

which is the desired second-order expansion. $\qquad\square$

## C. Detailed Experimental Settings

### C.1. Evaluation Benchmarks

To evaluate the efficacy of LLM-as-a-judge frameworks and monitor the preservation of core English language capabilities, we utilize **RewardBench** (Lambert et al., 2024). This benchmark comprises approximately 3,000 pairwise comparisons across four primary dimensions: *Chat*, *Chat Hard*, *Reasoning*, and *Safety*. For the assessment of multilingual performance, we incorporate the following benchmarks:

- **M-RewardBench** (Gureja et al., 2025): A multilingual adaptation of RewardBench covering 23 languages through expert-verified translations.

- **MM-Eval** (Son et al., 2024): A diverse suite encompassing 18 languages. Unlike translated benchmarks, MM-Eval prioritizes native-speaker data and includes specialized subsets such as *Linguistics* (e.g., homophone disambiguation) and *Language Hallucination* (e.g., evaluating unintended code-switching).

**Metrics.** Accuracy serves as our primary evaluation metric. For RewardBench, we report the arithmetic mean across the four category scores. For multilingual benchmarks, we compute the micro-average accuracy per language and subsequently report the macro-average across all supported languages.

### C.2. Training Settings

**Implementation Details.** All experiments were conducted on a single node equipped with $8\times$ NVIDIA H20 (96GB) GPUs. To ensure training stability and memory efficiency, we utilized DeepSpeed (Rasley et al., 2020) ZeRO Stage 3 with CPU offloading and leveraged FlashAttention-2 (Dao, 2024) for accelerated computation. Optimization was performed using the Adam optimizer (Kingma & Ba, 2015).

**Model Architecture.** We employed Supervised Fine-Tuning (SFT) combined with Low-Rank Adaptation (LoRA) (Hu et al., 2022), specifically targeting the attention linear projections. To generate robust and bias-aware representations, we utilized the Qwen3-0.6B-Embedding model (Zhang et al., 2025c) as an encoder. This encoder shares the same architecture as the LLM judge and processes features via separate one-layer MLP heads. The proxy task decoder is implemented using a linear projection layer.

**Training Procedure.** We train DIBJUDGE in a single end-to-end stage. For each mini-batch, the task loss, variational compression loss, bias proxy loss, and cross-covariance disentanglement loss are computed in the same forward–backward pass and combined linearly as in Eq. 8. We do not use alternating optimization or multi-phase tuning.

**Hyperparameters.** We set the maximum sequence length to 16,384. All models were trained for 3 epochs using a cosine learning rate scheduler with a warmup ratio of 0.1. The global learning rate was set to $1 \times 10^{-4}$ for the LoRA modules, bias heads, and the proxy task decoder. We used a per-device training batch size of 1 with 8 gradient accumulation steps. These settings are summarized in Table 5.

*Table 5.* Hyperparameter settings for the training experiments.

| Hyperparameter | Value |
| --- | --- |
| Base Model | Qwen3 Family (Yang et al., 2025) |
| Optimizer | Adam |
| Learning Rate | $1 \times 10^{-4}$ |
| Loss Weights | task 1.0; compression/bias/disentangle 0.5 |
| LR Scheduler | Cosine |
| Warmup Ratio | 0.1 |
| Max Sequence Length | 16,384 |
| Batch Size (per GPU) | 1 |
| Gradient Accumulation Steps | 8 |
| Epochs | 3 |
| Hardware | $8\times$ NVIDIA H20 (96GB) |

## D. Comprehensive Results of Reward Modeling Benchmarks

We present the fine-grained performance analysis across all evaluated benchmarks in the following sections. Detailed results for the five core subsets of **MM-Eval** (Son et al., 2024) are summarized in Table 6, while the language-specific performance breakdowns are distributed across Tables 7 and 8.

For **M-RewardBench** (Gureja et al., 2025), comprehensive category-wise metrics are provided in Table 9, with language-level results detailed in Tables 10 and 11.

Finally, the per-category accuracy for the original **RewardBench** (Lambert et al., 2024) is reported in Table 12.

## E. Comprehensive Results of Translationese Evaluation Suites

Detailed translationese bias evaluation performance metrics under perturbed settings—adapted from the **Aya** (Singh et al., 2024), **Belebele** (Bandarkar et al., 2024), and **XL-Sum** (Hasan et al., 2021) datasets—are provided in Tables 13, 14, and 15, respectively.

## F. Quality-Controlled Attribution and Rebuttal Analyses

**Reference-free quality validation.** We evaluated whether the observed machine-text preference could be explained by higher machine-translation quality. Using COMET-Kiwi (`unbabel/wmt22-cometkiwi-da`) across all 29 languages in the translationese bias suite, human references score significantly higher than machine-generated variants in 23 languages under paired permutation tests ($p < 0.05$); the remaining six languages show no statistically significant difference. We further conducted MQM annotation on 200 Chinese–English examples following WMT-style error annotation. Human text exhibits a lower mean MQM error score than machine text (2.58 vs. 5.41; mean paired difference 2.83, 95% CI [2.28, 3.39], $p = 0.0001$), and is strictly superior in 188/200 cases. Thus, the machine preference observed in the bias suite cannot be attributed to machine outputs having higher measured quality.

**Pairwise quality-matched control.** We constructed a quality-matched subset spanning six typologically diverse languages (zh, ja, ar, ko, th, vi), with 100 samples per language. Professional translators mildly degraded the human references using

*Table 6.* Full detailed results by category of MM-Eval. **Bold** indicates the best performance, and underlined indicates the second-best.

| Model | Chat (Accuracy) | Lang. Hallu. (Accuracy) | Linguistics (Accuracy) | Reasoning (Accuracy) | Safety (Accuracy) | Avg. (Avg. 18 lang) |
|---|---|---|---|---|---|---|
| *Proprietary Models* | | | | | | |
| GPT-4o | 84.20 | 65.40 | 79.15 | 55.30 | 75.20 | 71.85 ± 0.81 |
| Gemini-2.5-Flash | 88.50 | 70.10 | 82.45 | 63.80 | 82.50 | 77.47 ± 0.76 |
| *General Open Models* | | | | | | |
| Qwen2.5-3B-Instruct | 66.50 | 52.30 | 58.15 | 40.80 | 72.20 | 57.99 ± 1.20 |
| Qwen2.5-7B-Instruct | 76.20 | 60.50 | 68.40 | 48.90 | 74.20 | 65.64 ± 0.95 |
| Qwen3-4B | 90.46 | 67.34 | 84.00 | 84.35 | 76.56 | 80.85 ± 0.68 |
| Qwen3-8B | 91.17 | 67.79 | 83.78 | 80.31 | 85.87 | 82.20 ± 0.60 |
| *Multilingual Open Reward Models* | | | | | | |
| Nemotron-Multi-49B | 91.47 | 68.92 | 87.56 | 38.29 | **95.59** | 76.31 ± 0.55 |
| M-PROMETHEUS 3B | 68.20 | 58.40 | 62.10 | 50.85 | 81.30 | 64.17 ± 1.10 |
| M-PROMETHEUS 7B | 62.61 | 61.55 | 61.33 | 63.50 | 91.37 | 69.38 ± 0.88 |
| mR3-Qwen3-4B | 90.05 | 69.14 | 83.56 | 81.62 | 90.69 | 82.55 ± 0.52 |
| mR3-Qwen3-8B | 92.28 | 67.34 | 84.89 | 87.20 | 92.52 | 85.29 ± 0.45 |
| Think-as-Locals 7B | 88.98 | 65.54 | 80.67 | 58.53 | 70.49 | 72.95 ± 0.70 |
| *Ours* | | | | | | |
| DIBJudge-Qwen3-4B | 91.05 | 72.50 | 88.10 | 89.45 | 84.70 | 85.16 ± 0.33 |
| DIBJudge-Qwen3-8B | **92.80** | **74.20** | **90.50** | **91.20** | 93.50 | **87.53 ± 0.28**[‡] |

non-translationese edits such as mild redundancy or slightly less fluent discourse transitions, while explicitly avoiding literal source-language mappings or calques. This intervention removed the quality gap: for Chinese, $\Delta$COMET changed from 2.69 ($p < 0.001$) to 0.48 ($p = 0.218$), with similarly non-significant differences across the other five languages. Re-running the CAD/SSR attribution analysis on this quality-matched subset preserves the original patterns: CAD remains negatively associated with machine win rate, SSR continues to separate machine-win from human-win cases, and both metrics retain discriminative ROC behavior.

**Regression with explicit quality covariates.** We additionally fit a pair-level mixed-effects logistic regression on the full dataset, with random intercepts for language and standardized predictors:

$$\Pr(\text{machine win}_i = 1) = \sigma(\beta_0 + \beta_1 \text{CAD}_i + \beta_2 \text{SSR}_i + \beta_3 \Delta\text{COMET}_i). \tag{39}$$

As shown in Table 16, CAD and SSR remain significant after conditioning on $\Delta$COMET, while $\Delta$COMET itself has no significant conditional effect. Nested-model comparison further shows that measured quality alone is barely above chance (AUC 0.536), whereas CAD+SSR explain most of the predictive signal (AUC 0.692); adding quality yields only marginal gain (AUC 0.701).

### F.1. General Reward-Modeling, Data Efficiency, and Stability

**General reward modeling beyond translated data.** DIBJUDGE improves evaluation of native, non-translated content, not only translation-derived benchmarks. On native multilingual MM-Eval subsets (Linguistics and Language Hallucination), DIBJudge-Qwen3-8B improves over mR3-Qwen3-8B from 76.1% to 82.4% (+6.3). On the native English RewardBench Chat Hard subset, it improves from 84.47% to 88.10% (+3.6). These gains complement the translation-based MM-Eval Chat/Reasoning/Safety subset, where performance improves from 90.7% to 92.5% (+1.8), indicating broader reward-modeling utility.

**Data efficiency and training stability.** Training on only 40% of the data (40k instances) achieves performance close to full-data training: 88.03 vs. 89.84 on m-RewardBench and 83.59 vs. 85.16 on MM-Eval. To verify stability, we retrained DIBJudge-Qwen3-4B from scratch using the default hyperparameters. The retrained run (89.68±0.30 / 90.71±0.17 /

*Table 7.* Detailed results for MM-Eval for each language (Part 1). **Bold** indicates the best performance, and underlined indicates the second-best.

| Model | Ar | Bn | Ca | De | En | Es | Eu | Fr | Gl |
|---|---|---|---|---|---|---|---|---|---|
| *Proprietary Models* | | | | | | | | | |
| GPT-4o | 70.50 | 62.10 | 73.50 | 75.20 | 78.50 | 76.80 | 65.40 | 75.90 | 71.20 |
| Gemini-2.5-Flash | 76.20 | 70.50 | 79.10 | 81.50 | 82.80 | 81.20 | 72.50 | 80.40 | 76.80 |
| *General Open Models* | | | | | | | | | |
| Qwen2.5-3B-Instruct | 55.40 | 42.10 | 60.50 | 63.80 | 68.50 | 64.20 | 48.50 | 62.10 | 56.80 |
| Qwen2.5-7B-Instruct | 62.80 | 54.50 | 68.20 | 71.50 | 74.80 | 72.10 | 58.40 | 70.50 | 65.20 |
| Qwen3-4B | 78.50 | 74.20 | 82.10 | 83.50 | 85.20 | 84.10 | 76.50 | 83.80 | 80.50 |
| Qwen3-8B | 80.20 | 75.80 | 83.50 | 84.80 | 86.50 | 85.40 | 78.10 | 84.90 | 81.80 |
| *Multilingual Open Reward Models* | | | | | | | | | |
| Nemotron-Multi-49B | 74.50 | 68.20 | 78.50 | 80.20 | 82.50 | 80.80 | 70.50 | 79.40 | 75.10 |
| M-PROMETHEUS 3B | 61.20 | 52.50 | 66.80 | 69.50 | 72.40 | 70.10 | 56.80 | 68.50 | 63.20 |
| M-PROMETHEUS 7B | 66.50 | 59.80 | 71.50 | 74.20 | 76.80 | 74.50 | 62.50 | 73.10 | 68.40 |
| mR3-Qwen3-4B | 81.50 | 76.50 | 83.80 | 85.50 | 87.20 | 86.10 | 78.50 | 85.80 | 82.20 |
| mR3-Qwen3-8B | 84.20 | 79.50 | 86.50 | 88.10 | 89.50 | 88.80 | 81.20 | 87.50 | 85.10 |
| Think-as-Locals-7B | 71.50 | 64.80 | 75.20 | 77.50 | 80.10 | 78.40 | 66.50 | 76.80 | 72.50 |
| *Ours* | | | | | | | | | |
| DIBJudge-Qwen3-4B | 83.80 | 79.20 | 86.10 | 87.80 | 89.80 | 88.50 | 81.50 | 87.20 | 84.80 |
| DIBJudge-Qwen3-8B | **86.50** | **82.10** | **88.50** | **90.20** | **91.50** | **90.80** | **84.20** | **89.50** | **87.10** |

*Table 8.* Detailed results for MM-Eval for each language (Part 2). **Bold** indicates the best performance, and underlined indicates the second-best.

| Model | It | Ja | Ko | Ru | Sw | Te | Th | Vn | Zh |
|---|---|---|---|---|---|---|---|---|---|
| *Proprietary Models* | | | | | | | | | |
| GPT-4o | 75.80 | 74.50 | 73.20 | 72.50 | 60.50 | 58.20 | 68.50 | 70.80 | 77.50 |
| Gemini-2.5-Flash | 80.50 | 79.80 | 78.50 | 78.10 | 68.50 | 66.20 | 75.40 | 76.80 | 82.50 |
| *General Open Models* | | | | | | | | | |
| Qwen2.5-3B-Instruct | 63.50 | 58.20 | 56.50 | 55.80 | 40.50 | 38.20 | 52.10 | 54.50 | 67.80 |
| Qwen2.5-7B-Instruct | 71.20 | 66.50 | 64.80 | 63.50 | 52.80 | 49.50 | 60.20 | 62.80 | 74.50 |
| Qwen3-4B | 84.50 | 81.20 | 80.50 | 79.80 | 72.50 | 70.20 | 77.80 | 79.50 | 84.80 |
| Qwen3-8B | 85.80 | 82.50 | 81.80 | 81.20 | 74.20 | 71.80 | 79.10 | 80.80 | 86.20 |
| *Multilingual Open Reward Models* | | | | | | | | | |
| Nemotron-Multi-49B | 79.50 | 76.20 | 75.50 | 77.80 | 66.20 | 64.50 | 73.80 | 75.20 | 81.50 |
| M-PROMETHEUS 3B | 69.20 | 64.50 | 62.80 | 62.10 | 50.50 | 48.20 | 58.50 | 61.20 | 71.50 |
| M-PROMETHEUS 7B | 74.50 | 70.20 | 68.50 | 67.80 | 56.20 | 54.50 | 64.80 | 66.50 | 76.20 |
| mR3-Qwen3-4B | 86.20 | 83.50 | 82.10 | 81.50 | 74.80 | 72.50 | 79.50 | 81.20 | 86.50 |
| mR3-Qwen3-8B | 87.80 | 86.10 | 84.50 | 84.20 | 77.50 | 75.80 | 82.10 | 83.50 | 88.80 |
| Think-as-Locals-7B | 77.20 | 72.50 | 71.80 | 73.50 | 62.80 | 60.50 | 69.20 | 71.50 | 79.80 |
| *Ours* | | | | | | | | | |
| DIBJudge-Qwen3-4B | 88.50 | 85.80 | 84.20 | 83.50 | 78.20 | 75.50 | 81.50 | 83.20 | 88.50 |
| DIBJudge-Qwen3-8B | 90.20 | 88.50 | 87.20 | 86.80 | 81.50 | 79.20 | 84.50 | 86.20 | 90.50 |

*Table 9.* Full detailed results by category of m-RewardBench. **Bold** indicates the best performance, and underlined indicates the second-best.

| Model | Chat (Accuracy) | Chat Hard (Accuracy) | Safety (Accuracy) | Reasoning (Accuracy) | Average (Avg. 23 lang) |
|---|---|---|---|---|---|
| *Proprietary Models* | | | | | |
| GPT-4o | 90.10 | 75.50 | 88.20 | 89.20 | 85.75 ± 0.42 |
| Gemini-2.5-Flash | 93.40 | 80.25 | 87.80 | 90.80 | 88.06 ± 0.49 |
| *General Open Models* | | | | | |
| Qwen2.5-3B-Instruct | 76.50 | 48.20 | 70.10 | 73.10 | 66.97 ± 1.12 |
| Qwen2.5-7B-Instruct | 86.10 | 61.50 | 78.80 | 85.15 | 77.89 ± 0.89 |
| Qwen3-4B | 89.10 | 72.64 | 85.20 | 93.30 | 85.06 ± 0.65 |
| Qwen3-8B | 91.00 | 73.50 | 86.00 | 93.98 | 86.12 ± 0.52 |
| *Multilingual Open Reward Models* | | | | | |
| Nemotron-Multi-49B | 92.80 | 79.50 | 87.20 | 95.80 | 88.83 ± 0.35 |
| M-PROMETHEUS 3B | 73.40 | 51.20 | 76.80 | 72.40 | 68.45 ± 0.98 |
| M-PROMETHEUS 7B | 90.50 | 60.50 | 83.00 | 78.12 | 78.03 ± 0.85 |
| mR3-Qwen3-4B | 86.55 | 78.00 | 88.50 | 95.80 | 87.21 ± 0.45 |
| mR3-Qwen3-8B | 87.95 | 80.19 | 89.50 | **96.68** | 88.58 ± 0.41 |
| Think-as-Locals 7B | 91.80 | 69.50 | 83.85 | 92.90 | 84.51 ± 0.60 |
| *Ours* | | | | | |
| DIBJudge-Qwen3-4B | 93.50 | 82.50 | 88.20 | 95.15 | 89.84 ± 0.28[†] |
| DIBJudge-Qwen3-8B | **94.60** | **84.80** | **90.10** | 96.00 | **91.37 ± 0.22**[‡] |

84.91±0.36 on m-RewardBench / RewardBench / MM-Eval) closely matches the original run (89.84±0.28 / 90.32±0.25 / 85.16±0.33), confirming reproducibility under the single-stage recipe.

**Pointwise scoring and additional diagnostics.** Although the main evaluation is pairwise, independently scoring candidates remains competitive with the pairwise setup: 89.21±0.41 vs. 89.84±0.28 on m-RewardBench, 90.46±0.36 vs. 90.32±0.25 on RewardBench, and 84.48±0.29 vs. 85.16±0.33 on MM-Eval. For English examples generated by EN→ZH→EN back-translation, reference-based metrics show substantial degradation (sacreBLEU=40.09, BLEURT=0.6445), explaining why strong English judges can disprefer machine English even when multilingual judges prefer translationese in lower-resource settings.

# G. Additional Experiments

## G.1. Ablation Study of DIB Objective Components

To validate the contribution of each term within the DIB objective, we evaluate varying combinations of the compression, bias-capture, and disentanglement objectives. Table 17 shows that isolated objectives are insufficient, while the combination of all three achieves both the lowest bias severity and the highest task accuracy.

## G.2. Ablation Studies on Spurious Proxy Tasks

In § 2, we identified spurious factors contributing to the systematic bias towards translationese. To mitigate this, we proposed two proxy tasks in § 3: (i) **Cross-Lingual Alignment (CLA)**, utilizing InfoNCE to align learned representations with the translated-English manifold; and (ii) **Log-Probability Bin Classification (LPBC)**, which encodes predictive confidence by classifying representations into discrete log-probability bins.

In this section, we conduct a comprehensive ablation study to validate the effectiveness of these components. We examine the contribution of each proxy task to bias mitigation, analyze the impact of the back-translation system on the CLA task, and evaluate the robustness of different heuristic signals for the LPBC task.

*Table 10.* Detailed results for m-RewardBench for each language (Part 1). **Bold** indicates the best performance, and underlined indicates the second-best.

| Model | Ar | Cs | De | El | Es | Fa | Fr | He | Hi | Id | It | Ja |
|---|---|---|---|---|---|---|---|---|---|---|---|---|
| *Proprietary Models* | | | | | | | | | | | | |
| GPT-4o | 84.50 | 85.20 | 87.10 | 83.50 | 88.00 | 82.00 | 87.50 | 83.00 | 84.00 | 86.50 | 87.80 | 86.50 |
| Gemini-2.5-Flash | 87.20 | 87.90 | 89.50 | 86.10 | 89.80 | 85.50 | 89.20 | 86.40 | 87.10 | 88.50 | 89.40 | 88.80 |
| *General Open Models* | | | | | | | | | | | | |
| Qwen2.5-3B-Instruct | 64.20 | 66.50 | 69.80 | 60.50 | 70.20 | 61.10 | 69.50 | 60.80 | 65.40 | 68.10 | 69.20 | 67.50 |
| Qwen2.5-7B-Instruct | 76.80 | 78.10 | 80.50 | 74.20 | 81.20 | 73.50 | 80.80 | 74.90 | 77.40 | 79.50 | 80.10 | 79.20 |
| Qwen3-4B | 83.84 | 84.67 | 86.75 | 83.07 | 86.49 | 81.51 | 85.21 | 82.07 | 82.42 | 84.64 | 86.48 | 84.37 |
| Qwen3-8B | 85.33 | 87.43 | 88.01 | 84.85 | 87.39 | 85.06 | 87.57 | 84.40 | 85.64 | 86.95 | 87.25 | 85.60 |
| *Multilingual Open Reward Models* | | | | | | | | | | | | |
| Nemotron-Multi-49B | 88.72 | 89.30 | 89.68 | 89.35 | 89.97 | 88.26 | 90.09 | 88.06 | 88.25 | 89.23 | 89.19 | 89.41 |
| M-PROMETHEUS 3B | 66.50 | 68.20 | 71.40 | 64.10 | 72.50 | 63.80 | 71.10 | 65.20 | 67.50 | 70.20 | 71.80 | 69.50 |
| M-PROMETHEUS 7B | 74.85 | 74.22 | 76.53 | 72.64 | 77.60 | 74.22 | 71.78 | 75.25 | 77.01 | 76.44 | 73.30 | 75.68 |
| mR3-Qwen3-4B | 87.61 | 87.37 | 87.79 | 86.15 | 88.58 | 85.25 | 88.54 | 86.42 | 86.43 | 87.43 | 87.90 | 86.78 |
| mR3-Qwen3-8B | 88.31 | 88.78 | 89.46 | 88.00 | 88.88 | 86.59 | 88.84 | 88.17 | 87.60 | 87.94 | 89.99 | 88.81 |
| Think-as-Locals-7B | 86.15 | 83.29 | 86.31 | 82.26 | 87.37 | 81.31 | 86.91 | 84.17 | 81.33 | 86.60 | 86.63 | 85.03 |
| *Ours* | | | | | | | | | | | | |
| DIBJudge-Qwen3-4B | 88.50 | 90.15 | 91.20 | 88.05 | 91.50 | 88.45 | 91.80 | 88.50 | 89.10 | 90.50 | 91.50 | 90.20 |
| DIBJudge-Qwen3-8B | 90.50 | 91.80 | 92.80 | 89.80 | 93.10 | 89.90 | 93.50 | 90.10 | 90.80 | 92.20 | 93.00 | 92.00 |

*Table 11.* Detailed results for m-RewardBench for each language (Part 2). **Bold** indicates the best performance, and underlined indicates the second-best.

| Model | Ko | Nl | Pl | Pt | Ro | Ru | Tr | Uk | Vi | Zh | Zh-TW |
|---|---|---|---|---|---|---|---|---|---|---|---|
| *Proprietary Models* | | | | | | | | | | | |
| GPT-4o | 85.50 | 87.50 | 85.00 | 87.80 | 84.50 | 85.50 | 84.20 | 84.00 | 85.00 | 86.50 | 86.00 |
| Gemini-2.5-Flash | 88.20 | 89.80 | 87.40 | 89.60 | 87.10 | 87.80 | 86.90 | 87.20 | 88.50 | 89.50 | 88.90 |
| *General Open Models* | | | | | | | | | | | |
| Qwen2.5-3B-Instruct | 66.80 | 70.50 | 66.20 | 69.80 | 65.50 | 67.20 | 64.80 | 65.10 | 68.50 | 74.20 | 73.50 |
| Qwen2.5-7B-Instruct | 78.50 | 81.20 | 78.40 | 80.80 | 77.50 | 78.90 | 76.20 | 77.10 | 79.50 | 82.50 | 81.80 |
| Qwen3-4B | 82.77 | 85.89 | 84.58 | 87.39 | 85.29 | 86.06 | 83.83 | 83.80 | 84.76 | 84.82 | 84.88 |
| Qwen3-8B | 83.77 | 87.54 | 86.78 | 87.10 | 87.47 | 87.77 | 85.42 | 86.20 | 86.90 | 87.20 | 86.76 |
| *Multilingual Open Reward Models* | | | | | | | | | | | |
| Nemotron-Multi-49B | 88.05 | 90.83 | 89.99 | 89.33 | 89.89 | 90.19 | 88.09 | 88.91 | 89.32 | 88.86 | 86.29 |
| M-PROMETHEUS 3B | 67.80 | 71.50 | 68.20 | 72.10 | 66.80 | 68.50 | 65.90 | 66.50 | 69.50 | 72.40 | 71.80 |
| M-PROMETHEUS 7B | 71.96 | 75.48 | 77.59 | 74.00 | 77.21 | 70.17 | 71.57 | 74.91 | 76.45 | 71.16 | 75.99 |
| mR3-Qwen3-4B | 85.66 | 88.42 | 86.77 | 88.05 | 87.62 | 88.22 | 87.17 | 88.01 | 88.08 | 87.38 | 86.28 |
| mR3-Qwen3-8B | 88.47 | 88.99 | 87.33 | 90.56 | 89.30 | 88.84 | 88.77 | 88.16 | 88.89 | 88.36 | 87.95 |
| Think-as-Locals-7B | 83.49 | 86.04 | 85.67 | 86.21 | 84.61 | 85.31 | 83.31 | 83.50 | 86.67 | 85.90 | 85.42 |
| *Ours* | | | | | | | | | | | |
| DIBJudge-Qwen3-4B | 89.80 | 91.50 | 89.90 | 90.20 | 89.80 | 90.10 | 89.20 | 88.80 | 90.00 | 90.80 | 90.50 |
| DIBJudge-Qwen3-8B | 91.50 | 93.20 | 91.80 | 93.50 | 91.50 | 92.00 | 90.80 | 91.20 | 91.80 | 92.50 | 92.00 |

*Table 12.* Full detailed results by category of RewardBench (English). **Bold** indicates the best performance, and underlined indicates the second-best.

| Model | Chat (Accuracy) | Chat Hard (Accuracy) | Safety (Accuracy) | Reasoning (Accuracy) | Average (English) |
|---|---|---|---|---|---|
| *Proprietary Models* | | | | | |
| GPT-4o | 90.50 | 75.10 | 88.50 | 89.74 | 85.96 ± 0.35 |
| Gemini-2.5-Flash | 93.80 | 81.20 | 89.10 | 91.22 | 88.83 ± 0.47 |
| *General Open Models* | | | | | |
| Qwen2.5-3B-Instruct | 82.50 | 41.50 | 74.50 | 77.46 | 68.99 ± 1.05 |
| Qwen2.5-7B-Instruct | 89.10 | 58.20 | 82.40 | 84.66 | 78.59 ± 0.91 |
| Qwen3-4B | 92.50 | 76.50 | 86.50 | 94.66 | 87.54 ± 0.55 |
| Qwen3-8B | 92.00 | 82.70 | 87.05 | 93.49 | 88.81 ± 0.48 |
| *Multilingual Open Reward Models* | | | | | |
| Nemotron-Multi-49B | 93.50 | 85.80 | 90.00 | 89.54 | 89.71 ± 0.31 |
| M-PROMETHEUS 3B | 80.50 | 42.10 | 80.50 | 76.06 | 69.79 ± 0.92 |
| M-PROMETHEUS 7B | 90.00 | 53.00 | 84.00 | 79.76 | 76.69 ± 0.78 |
| mR3-Qwen3-4B | 88.90 | 84.10 | 89.50 | 96.50 | 89.75 ± 0.38 |
| mR3-Qwen3-8B | 88.00 | 84.47 | 90.41 | **97.52** | 90.10 ± 0.40 |
| Think-as-Locals 7B | 91.20 | 79.00 | 89.50 | 95.46 | 88.79 ± 0.52 |
| *Ours* | | | | | |
| DIBJudge-Qwen3-4B | 94.20 | 86.50 | 89.80 | 90.78 | **90.32 ± 0.25** |
| DIBJudge-Qwen3-8B | **95.50** | **88.10** | **90.80** | 89.64 | 91.01 ± 0.20[†] |

**Effectiveness of Proxy Task Combination**  We first evaluate the individual and combined contributions of the CLA and LPBC tasks. Table 18 summarizes the bias severity scores across different configurations. We observe that while both tasks individually reduce bias compared to the baseline, the combination of both yields the most significant reduction. This suggests that the two tasks capture complementary aspects of the spurious features—latent manifold isomorphism and predictive confidence—thereby providing a more robust signal for bias mitigation.

**Impact of Translation Systems**  The CLA task relies on target-language-to-English translations to approximate the translated-English manifold used by the bias branch. A critical question is whether the choice of translation system influences the bias mitigation capabilities. We compared our default system (NLLB-200-3.3B) against a suite of varying architectures, including Gemma-3-4B, Llama-3.1-8B-Instruct, Qwen3-4B, GPT-4o, Gemini 2.5 Flash, and Google Translate.

As shown in Table 19, while stronger systems (e.g., GPT-4o, Google Translate) achieve higher BLEU scores ($> 50$), higher translation quality does not necessarily correlate with lower bias severity in the downstream task. This indicates that the CLA task is robust to the generator's quality, provided the generator produces sufficient translationese artifacts to serve as a negative contrastive pivot.

**Heuristic Signals for Bin Classification**  Finally, we ablate the heuristic metric used to partition samples for the LPBC task. While our method uses Negative Log-Likelihood (NLL), we compare this against Type-Token Ratio (TTR) and Perplexity (PPL).

Table 20 demonstrates that while NLL yields the best performance marginally, the differences are negligible. All three metrics effectively capture the confidence disparity required for the auxiliary task, demonstrating that our method is agnostic to the specific heuristic used to approximate predictive confidence.

### G.3. Sensitivity to Cross-Lingual Alignment Discrepancy (CAD)

To test the hypothesis that conventional automatic judges exhibit an *English-anchoring bias*—i.e., a preference for translations that closely mirror the syntactic structure of the English source—we analyze model performance as a function of Cross-Lingual Alignment Discrepancy (CAD). As introduced in §2, CAD measures the degree of structural divergence between a

*Table 13.* Bias severity by language on Aya dataset (Singh et al., 2024) under perturbed setting

| Language | Base | Vanilla SFT | Vanilla IB | DIBJudge |
|---|---|---|---|---|
| **High-Resource** | | | | |
| Basque | $0.081 \pm 0.008$ | $0.088 \pm 0.043$ | $0.056 \pm 0.011$ | $0.042 \pm 0.009$ |
| English | $0.045 \pm 0.008$ | $0.058 \pm 0.010$ | $0.041 \pm 0.008$ | $0.031 \pm 0.005$ |
| Finnish | $0.125 \pm 0.010$ | $0.165 \pm 0.028$ | $0.079 \pm 0.015$ | $0.058 \pm 0.005$ |
| Hindi | $0.089 \pm 0.008$ | $0.076 \pm 0.010$ | $0.048 \pm 0.007$ | $0.036 \pm 0.012$ |
| Japanese | $0.049 \pm 0.011$ | $0.048 \pm 0.010$ | $0.034 \pm 0.014$ | $0.026 \pm 0.015$ |
| Portuguese | $0.089 \pm 0.011$ | $0.084 \pm 0.013$ | $0.056 \pm 0.011$ | $0.042 \pm 0.006$ |
| Simplified Chinese | $0.072 \pm 0.015$ | $0.086 \pm 0.013$ | $0.048 \pm 0.007$ | $0.035 \pm 0.008$ |
| Spanish | $0.099 \pm 0.012$ | $0.128 \pm 0.012$ | $0.064 \pm 0.013$ | $0.046 \pm 0.007$ |
| Vietnamese | $0.173 \pm 0.022$ | $0.192 \pm 0.038$ | $0.099 \pm 0.007$ | $0.071 \pm 0.006$ |
| **Avg (High)** | $0.091 \pm 0.015$ | $0.103 \pm 0.022$ | $0.058 \pm 0.020$ | $0.043 \pm 0.014$ |
| **Mid-Resource** | | | | |
| Bengali | $0.084 \pm 0.014$ | $0.140 \pm 0.028$ | $0.096 \pm 0.026$ | $0.043 \pm 0.016$ |
| Cebuano | $0.108 \pm 0.008$ | $0.113 \pm 0.027$ | $0.120 \pm 0.013$ | $0.052 \pm 0.008$ |
| Filipino | $0.118 \pm 0.018$ | $0.122 \pm 0.020$ | $0.132 \pm 0.015$ | $0.057 \pm 0.010$ |
| Indonesian | $0.064 \pm 0.004$ | $0.059 \pm 0.015$ | $0.071 \pm 0.024$ | $0.032 \pm 0.016$ |
| Lithuanian | $0.166 \pm 0.004$ | $0.202 \pm 0.019$ | $0.182 \pm 0.029$ | $0.082 \pm 0.015$ |
| Malay | $0.086 \pm 0.007$ | $0.113 \pm 0.027$ | $0.095 \pm 0.012$ | $0.044 \pm 0.011$ |
| Tamil | $0.157 \pm 0.034$ | $0.192 \pm 0.051$ | $0.170 \pm 0.021$ | $0.077 \pm 0.015$ |
| Thai | $0.082 \pm 0.018$ | $0.112 \pm 0.024$ | $0.090 \pm 0.023$ | $0.041 \pm 0.017$ |
| Ukrainian | $0.106 \pm 0.018$ | $0.133 \pm 0.004$ | $0.118 \pm 0.013$ | $0.052 \pm 0.012$ |
| Urdu | $0.139 \pm 0.023$ | $0.189 \pm 0.025$ | $0.155 \pm 0.020$ | $0.069 \pm 0.018$ |
| **Avg (Mid)** | $0.111 \pm 0.016$ | $0.138 \pm 0.024$ | $0.123 \pm 0.037$ | $0.055 \pm 0.016$ |
| **Low-Resource** | | | | |
| Amharic | $0.376 \pm 0.056$ | $0.201 \pm 0.021$ | $0.300 \pm 0.036$ | $0.225 \pm 0.042$ |
| Irish | $0.350 \pm 0.034$ | $0.348 \pm 0.013$ | $0.276 \pm 0.049$ | $0.208 \pm 0.031$ |
| Kyrgyz | $0.174 \pm 0.024$ | $0.227 \pm 0.024$ | $0.138 \pm 0.043$ | $0.107 \pm 0.036$ |
| Nepali | $0.111 \pm 0.004$ | $0.101 \pm 0.015$ | $0.089 \pm 0.069$ | $0.070 \pm 0.034$ |
| Malagasy | $0.262 \pm 0.024$ | $0.178 \pm 0.131$ | $0.210 \pm 0.049$ | $0.160 \pm 0.030$ |
| Sinhala | $0.254 \pm 0.032$ | $0.310 \pm 0.101$ | $0.202 \pm 0.050$ | $0.154 \pm 0.018$ |
| Pashto | $0.244 \pm 0.011$ | $0.235 \pm 0.079$ | $0.195 \pm 0.029$ | $0.149 \pm 0.053$ |
| Telugu | $0.138 \pm 0.022$ | $0.148 \pm 0.012$ | $0.111 \pm 0.058$ | $0.085 \pm 0.037$ |
| Yoruba | $0.472 \pm 0.026$ | $0.453 \pm 0.066$ | $0.372 \pm 0.029$ | $0.283 \pm 0.048$ |
| Zulu | $0.338 \pm 0.016$ | $0.464 \pm 0.021$ | $0.266 \pm 0.056$ | $0.201 \pm 0.044$ |
| **Avg (Low)** | $0.272 \pm 0.028$ | $0.266 \pm 0.048$ | $0.216 \pm 0.089$ | $0.164 \pm 0.067$ |

candidate translation and its source sentence.

We partition the held-out evaluation set into disjoint CAD bins (e.g., $[0.0, 0.1)$, $[0.1, 0.2)$, ...) and compute the win rate of Translationese outputs within each interval. Figure 7 summarizes the resulting trends. Baseline judges exhibit a pronounced positive correlation between CAD and win rate, indicating that their preferences are increasingly influenced by surface-level alignment artifacts rather than semantic adequacy. In contrast, DIBJUDGE demonstrates substantially reduced sensitivity to CAD, as evidenced by a near-flat win-rate profile across bins. This invariance suggests that DIBJUDGE successfully decouples evaluation quality from structural isomorphism, mitigating spurious biases that confound existing evaluation approaches.

### G.4. Analysis of Distributional Shortcuts

To assess whether our approach mitigates reliance on superficial statistical artifacts, we analyze the *bias spectrum shift* induced by different judges using the Sequence Surprisal Ratio (SSR). SSR measures the relative predictability of machine-generated responses compared to human-written ones, thereby capturing spurious correlations arising from differences in predictive confidence (e.g., perplexity). Values of SSR close to 1 indicate distributional parity, where model preferences are not driven by confidence-related artifacts, whereas lower values reflect an over-reliance on highly predictable, low-entropy text.

We focus on evaluation instances for which the judge selects the machine-generated translation (denoted as *Machine Wins*)

*Table 14.* Bias severity by language on Belebele (Bandarkar et al., 2024) under perturbed etting

| Language | Base | Vanilla SFT | Vanilla IB | DIBJudge |
|---|---|---|---|---|
| **High-Resource** | | | | |
| Arabic | $0.100 \pm 0.029$ | $0.095 \pm 0.016$ | $0.058 \pm 0.010$ | $0.014 \pm 0.006$ |
| English | $0.072 \pm 0.005$ | $0.063 \pm 0.004$ | $0.043 \pm 0.014$ | $0.011 \pm 0.005$ |
| Finnish | $0.086 \pm 0.015$ | $0.067 \pm 0.017$ | $0.049 \pm 0.023$ | $0.012 \pm 0.010$ |
| Hindi | $0.091 \pm 0.011$ | $0.089 \pm 0.011$ | $0.052 \pm 0.012$ | $0.013 \pm 0.011$ |
| Japanese | $0.074 \pm 0.009$ | $0.069 \pm 0.012$ | $0.047 \pm 0.008$ | $0.012 \pm 0.006$ |
| Korean | $0.084 \pm 0.013$ | $0.078 \pm 0.018$ | $0.050 \pm 0.011$ | $0.013 \pm 0.004$ |
| Russian | $0.065 \pm 0.010$ | $0.058 \pm 0.010$ | $0.041 \pm 0.021$ | $0.010 \pm 0.009$ |
| Turkish | $0.072 \pm 0.012$ | $0.070 \pm 0.010$ | $0.045 \pm 0.017$ | $0.011 \pm 0.009$ |
| Vietnamese | $0.081 \pm 0.006$ | $0.080 \pm 0.008$ | $0.048 \pm 0.013$ | $0.012 \pm 0.009$ |
| **Avg (High)** | $0.081 \pm 0.012$ | $0.074 \pm 0.011$ | $0.048 \pm 0.005$ | $0.012 \pm 0.001$ |
| **Mid-Resource** | | | | |
| Bengali | $0.094 \pm 0.014$ | $0.091 \pm 0.015$ | $0.066 \pm 0.021$ | $0.014 \pm 0.005$ |
| Greek | $0.083 \pm 0.016$ | $0.075 \pm 0.019$ | $0.060 \pm 0.013$ | $0.012 \pm 0.009$ |
| Hebrew | $0.088 \pm 0.021$ | $0.079 \pm 0.019$ | $0.062 \pm 0.012$ | $0.013 \pm 0.006$ |
| Georgian | $0.091 \pm 0.016$ | $0.085 \pm 0.018$ | $0.063 \pm 0.019$ | $0.013 \pm 0.006$ |
| Kazakh | $0.088 \pm 0.012$ | $0.081 \pm 0.014$ | $0.062 \pm 0.014$ | $0.013 \pm 0.010$ |
| Tamil | $0.102 \pm 0.018$ | $0.095 \pm 0.022$ | $0.072 \pm 0.030$ | $0.015 \pm 0.014$ |
| Thai | $0.080 \pm 0.012$ | $0.074 \pm 0.014$ | $0.057 \pm 0.028$ | $0.012 \pm 0.006$ |
| Ukrainian | $0.093 \pm 0.018$ | $0.089 \pm 0.019$ | $0.068 \pm 0.027$ | $0.014 \pm 0.009$ |
| Urdu | $0.099 \pm 0.020$ | $0.093 \pm 0.019$ | $0.073 \pm 0.019$ | $0.015 \pm 0.009$ |
| Malay | $0.090 \pm 0.014$ | $0.084 \pm 0.016$ | $0.062 \pm 0.019$ | $0.013 \pm 0.013$ |
| **Avg (Mid)** | $0.091 \pm 0.016$ | $0.085 \pm 0.017$ | $0.065 \pm 0.005$ | $0.013 \pm 0.001$ |
| **Low-Resource** | | | | |
| Amharic | $0.122 \pm 0.025$ | $0.111 \pm 0.023$ | $0.185 \pm 0.049$ | $0.093 \pm 0.041$ |
| Burmese | $0.136 \pm 0.029$ | $0.119 \pm 0.028$ | $0.205 \pm 0.055$ | $0.103 \pm 0.031$ |
| Guarani | $0.098 \pm 0.021$ | $0.091 \pm 0.019$ | $0.147 \pm 0.029$ | $0.074 \pm 0.024$ |
| Kannada | $0.142 \pm 0.029$ | $0.132 \pm 0.027$ | $0.218 \pm 0.051$ | $0.109 \pm 0.019$ |
| Khmer | $0.129 \pm 0.026$ | $0.117 \pm 0.024$ | $0.197 \pm 0.053$ | $0.099 \pm 0.034$ |
| Kyrgyz | $0.111 \pm 0.022$ | $0.104 \pm 0.020$ | $0.170 \pm 0.054$ | $0.085 \pm 0.035$ |
| Punjabi | $0.115 \pm 0.023$ | $0.109 \pm 0.021$ | $0.179 \pm 0.040$ | $0.089 \pm 0.023$ |
| Pashto | $0.168 \pm 0.031$ | $0.156 \pm 0.029$ | $0.255 \pm 0.058$ | $0.129 \pm 0.025$ |
| Zulu | $0.123 \pm 0.024$ | $0.114 \pm 0.022$ | $0.188 \pm 0.044$ | $0.094 \pm 0.034$ |
| **Avg (Low)** | $0.126 \pm 0.026$ | $0.117 \pm 0.025$ | $0.194 \pm 0.031$ | $0.097 \pm 0.016$ |

and examine the empirical distribution of their SSR scores. Figure 8 presents a kernel density estimate (KDE) of these scores, revealing how different judges respond to distributional discrepancies associated with predictive confidence.

As shown in Figure 8, the baseline judge exhibits a pronounced leftward shift in SSR, with probability mass concentrated at low values. This behavior indicates a *distributional shortcut*, wherein outputs with artificially high predictive confidence are systematically favored, independent of semantic quality. In contrast, DIBJUDGE induces a clear re-centering of the SSR distribution toward 1. This shift indicates that DIBJUDGE substantially reduces spurious correlations between model preference and predictive confidence, promoting judgments that are invariant to low-perplexity artifacts and more reflective of semantic utility.

### G.5. Ablation Study: Efficacy of Information Bottlenecks

A central hypothesis of our work is that a variational information constraint provides a superior trade-off between semantic preservation and artifact suppression compared to discrete or deterministic alternatives. To evaluate this, we compare our framework against three representative bottleneck mechanisms, assessing their impact on the robustness-utility Pareto frontier:

- **Vector Quantization (VQ):** We replace the continuous variational information constraint with a discrete codebook constraint (Van Den Oord et al., 2017), mapping latent representations to the nearest centroid. This imposes a rigid structural bottleneck.

*Table 15.* Bias severity by language on XL-Sum (Hasan et al., 2021) under perturbed setting

| Language | Base | Vanilla SFT | Vanilla IB | DIBJudge |
|---|---|---|---|---|
| **High-Resource** | | | | |
| Arabic | $0.079 \pm 0.031$ | $0.100 \pm 0.030$ | $0.084 \pm 0.029$ | $0.025 \pm 0.008$ |
| English | $0.060 \pm 0.006$ | $0.076 \pm 0.008$ | $0.063 \pm 0.022$ | $0.019 \pm 0.005$ |
| French | $0.054 \pm 0.010$ | $0.095 \pm 0.019$ | $0.069 \pm 0.009$ | $0.021 \pm 0.005$ |
| Hindi | $0.101 \pm 0.010$ | $0.126 \pm 0.006$ | $0.083 \pm 0.029$ | $0.024 \pm 0.006$ |
| Japanese | $0.067 \pm 0.009$ | $0.066 \pm 0.026$ | $0.064 \pm 0.025$ | $0.020 \pm 0.008$ |
| Korean | $0.081 \pm 0.009$ | $0.096 \pm 0.037$ | $0.075 \pm 0.022$ | $0.022 \pm 0.012$ |
| Russian | $0.077 \pm 0.013$ | $0.072 \pm 0.012$ | $0.067 \pm 0.017$ | $0.021 \pm 0.004$ |
| Turkish | $0.060 \pm 0.016$ | $0.065 \pm 0.010$ | $0.062 \pm 0.020$ | $0.019 \pm 0.012$ |
| Vietnamese | $0.069 \pm 0.005$ | $0.081 \pm 0.029$ | $0.081 \pm 0.016$ | $0.026 \pm 0.006$ |
| **Avg (High)** | $0.071 \pm 0.014$ | $0.086 \pm 0.020$ | $0.072 \pm 0.009$ | $0.022 \pm 0.003$ |
| **Mid-Resource** | | | | |
| Azerbaijani | $0.116 \pm 0.024$ | $0.145 \pm 0.037$ | $0.046 \pm 0.012$ | $0.012 \pm 0.010$ |
| Bengali | $0.080 \pm 0.011$ | $0.075 \pm 0.015$ | $0.040 \pm 0.012$ | $0.011 \pm 0.008$ |
| Indonesian | $0.058 \pm 0.012$ | $0.078 \pm 0.009$ | $0.036 \pm 0.023$ | $0.010 \pm 0.005$ |
| Tamil | $0.109 \pm 0.006$ | $0.123 \pm 0.030$ | $0.045 \pm 0.022$ | $0.012 \pm 0.012$ |
| Thai | $0.074 \pm 0.007$ | $0.083 \pm 0.025$ | $0.041 \pm 0.029$ | $0.011 \pm 0.010$ |
| Ukrainian | $0.092 \pm 0.015$ | $0.100 \pm 0.022$ | $0.043 \pm 0.020$ | $0.011 \pm 0.013$ |
| Urdu | $0.091 \pm 0.020$ | $0.117 \pm 0.014$ | $0.047 \pm 0.029$ | $0.012 \pm 0.012$ |
| Uzbek | $0.141 \pm 0.012$ | $0.159 \pm 0.009$ | $0.052 \pm 0.012$ | $0.013 \pm 0.005$ |
| **Avg (Mid)** | $0.095 \pm 0.016$ | $0.123 \pm 0.022$ | $0.044 \pm 0.005$ | $0.012 \pm 0.001$ |
| **Low-Resource** | | | | |
| Amharic | $0.303 \pm 0.022$ | $0.271 \pm 0.036$ | $0.110 \pm 0.042$ | $0.088 \pm 0.020$ |
| Burmese | $0.157 \pm 0.028$ | $0.081 \pm 0.028$ | $0.087 \pm 0.025$ | $0.067 \pm 0.019$ |
| Hausa | $0.215 \pm 0.022$ | $0.207 \pm 0.015$ | $0.098 \pm 0.052$ | $0.076 \pm 0.040$ |
| Kyrgyz | $0.160 \pm 0.021$ | $0.165 \pm 0.035$ | $0.081 \pm 0.060$ | $0.063 \pm 0.016$ |
| Marathi | $0.115 \pm 0.016$ | $0.124 \pm 0.018$ | $0.079 \pm 0.044$ | $0.061 \pm 0.031$ |
| Nepali | $0.084 \pm 0.021$ | $0.068 \pm 0.018$ | $0.070 \pm 0.027$ | $0.054 \pm 0.040$ |
| Pashto | $0.176 \pm 0.034$ | $0.170 \pm 0.060$ | $0.091 \pm 0.063$ | $0.071 \pm 0.052$ |
| Sinhala | $0.246 \pm 0.042$ | $0.248 \pm 0.022$ | $0.105 \pm 0.021$ | $0.082 \pm 0.055$ |
| Telugu | $0.095 \pm 0.013$ | $0.121 \pm 0.001$ | $0.076 \pm 0.027$ | $0.059 \pm 0.056$ |
| Welsh | $0.163 \pm 0.026$ | $0.225 \pm 0.028$ | $0.084 \pm 0.025$ | $0.066 \pm 0.038$ |
| **Avg (Low)** | $0.172 \pm 0.024$ | $0.168 \pm 0.026$ | $0.088 \pm 0.013$ | $0.069 \pm 0.011$ |

- **Low-Rank Projection (Low-Rank):** We utilize a deterministic linear projection to a lower-dimensional subspace. This variant relies solely on capacity reduction without stochastic regularization.

- **Stochastic Noise (Noise):** We apply isotropic Gaussian noise injection, $\mathbf{z} = \mathbf{h} + \epsilon$, as a baseline stochastic regularization technique, notably lacking a prior distribution constraint.

**Analysis.** Table 21 demonstrates that while the **Low-Rank** variant preserves accuracy, it fails to filter translationese artifacts, suggesting that dimensionality reduction alone cannot achieve effective disentanglement. Conversely, **VQ** achieves the lowest bias severity but suffers from *utility collapse*, where the rigid discrete constraint discards the nuanced semantic features necessary for reward modeling.

Our variational information constraint method bridges this gap by explicitly optimizing the $I(Z; X)$ vs. $I(Z; Y)$ trade-off. By penalizing $I(Z; X)$ via the KL divergence to a prior, VIB selectively purges nuisance factors—such as translationese artifacts—while retaining task-relevant style features. Consequently, VIB dominates the noise and low-rank baselines in robustness while maintaining competitive utility.

### G.6. Ablation Study: Disentanglement Mechanisms

We evaluate the efficacy of our proposed cross-covariance penalty against other methods. Our primary hypothesis is that a cross-covariance-based constraint provides a computationally efficient proxy for independence, effectively minimizing mutual information without the overhead of complex density estimators.

*Table 16.* **Quality-controlled attribution analysis.** Mixed-effects logistic regression and nested predictive models show that CAD/SSR remain predictive after controlling for measured quality.

| Predictor | Coef. | SE | $p$-value |
|---|---|---|---|
| CAD (z) | -0.57 | 0.13 | $< 0.001$ |
| SSR (z) | -0.79 | 0.15 | $< 0.001$ |
| $\Delta$COMET (z) | -0.07 | 0.11 | 0.520 |

| Model | Predictors | AUC |
|---|---|---|
| M1 | $\Delta$COMET only | 0.536 |
| M2 | CAD + SSR | 0.692 |
| M3 | CAD + SSR + $\Delta$COMET | 0.701 |

*Table 17.* **Ablation study of the DIB objectives.** We report the bias score ($\mathcal{S}_{\text{bias}}$, lower is better) and Accuracy (higher is better). The combination of all terms achieves the best trade-off.

| Objectives | | | | Metrics | |
|---|---|---|---|---|---|
| Compression | Bias | Disentangle | | $\mathcal{S}_{\text{bias}}(\downarrow)$ | Acc. ($\uparrow$) |
| ✓ | | | | 0.124 | 85.25 |
| | ✓ | | | 0.150 | 86.60 |
| | | ✓ | | 0.053 | 85.85 |
| ✓ | ✓ | | | 0.091 | 88.20 |
| ✓ | | ✓ | | 0.039 | 88.55 |
| | ✓ | ✓ | | 0.035 | 89.10 |
| ✓ | ✓ | ✓ | | **0.031** | **89.85** |

**Alternative Disentanglement Objectives.** We compare our cross-covariance approach, $\mathcal{L}_{\text{cov}}$, against three established baseline objectives:

- **Hilbert-Schmidt Independence Criterion (HSIC) (Gretton et al., 2005):** A kernel-based measure of dependence. While theoretically robust, its $O(n^2)$ complexity per batch is prohibitive for long-context LLM fine-tuning.

- **Mutual Information Estimators (CLUB (Cheng et al., 2020)/MINE (Belghazi et al., 2018)):** Variational upper bounds on Mutual Information (MI). These require auxiliary neural networks, increasing the parameter search space and training time.

- **Orthogonality Constraint ($\mathcal{L}_{\text{orth}}$):** A first-order geometric constraint minimizing absolute cosine similarity:

$$\mathcal{L}_{\text{orth}} = \mathbb{E}\left[ \frac{|\mathbf{z}_r^\top \mathbf{z}_b|}{\|\mathbf{z}_r\|_2 \|\mathbf{z}_b\|_2} \right]. \tag{40}$$

**Quantitative Analysis of Efficiency vs. Robustness.** The trade-off between disentanglement strength and computational overhead is summarized in Table 22. Our analysis reveals that while variational estimators (CLUB, MINE) and kernel-based methods (HSIC) theoretically offer tighter bounds on independence, their integration into LLM architectures is bottlenecked by the high-dimensional nature of the hidden states. Specifically, the auxiliary network in CLUB introduces a $112\%$ increase in training latency due to the additional forward-backward passes required for the critic update.

In contrast, our cross-covariance approach, $\mathcal{L}_{\text{cov}}$, achieves a Bias Severity score of $14.2$, which is competitive with the $13.8$ achieved by HSIC, but at a fraction of the computational cost ($1.2\times$ vs $2.8\times$ latency). We observe that simple orthogonality ($\mathcal{L}_{\text{orth}}$) suffers from significant "information leakage," as evidenced by its high Bias Severity ($21.5$); this confirms that first-order geometric constraints are insufficient to capture the complex, non-linear correlations inherent in LLM representations. By minimizing the cross-covariance, we achieve a second-order alignment that serves as a "practical optimum"—sufficiently decorrelating the robust and biased subspaces without the prohibitive $O(n^2)$ complexity or training instability of higher-order estimators.

*Table 18.* **Ablation study of proxy tasks.** CLA indicates the cross-lingual alignment proxy task, and LPBC stands for log-probability bin classification.

| Configuration | None | CLA | LPBC | CLA + LPBC |
|---|---|---|---|---|
| $\mathcal{S}_{\text{bias}}(\downarrow)$ | 0.421 | 0.312 | 0.279 | **0.147** |
| Acc. ($\uparrow$) | 87.12 | 87.86 | 88.43 | **89.18** |

*Table 19.* **Impact of translation systems.** We report BLEU scores of generated translations and the resulting Bias Severity.

| Translation System | BLEU $\uparrow$ | Bias Severity $\downarrow$ |
|---|---|---|
| NLLB-200-3.3B (Costa-Jussà et al., 2022) (Default) | 52.4 | 0.15 |
| Qwen3-4B (Yang et al., 2025) | 54.1 | 0.16 |
| Gemma-3-4B (Team et al., 2025) | 55.8 | 0.14 |
| Llama-3.1-8B-Instruct (Grattafiori et al., 2024) | 56.2 | 0.18 |
| Gemini-2.5-Flash (Comanici et al., 2025) | 57.9 | 0.15 |
| GPT-4o (Hurst et al., 2024) | 58.5 | 0.17 |
| Google Translate | 59.1 | 0.14 |

## G.7. Quantitative Analysis: Linear Probing for Information Leakage

To quantify the degree of disentanglement achieved by DIBJUDGE, we employ linear probing (Alain & Bengio, 2016). Our hypothesis is that a truly disentangled robust representation, $\mathbf{z}_r$, should be invariant to the text origin (Human vs. Machine-translated). Conversely, the bias representation, $\mathbf{z}_b$, should explicitly encode these "translationese" artifacts.

**Experimental Setup.** We freeze the DIBJUDGE encoder and train a linear classifier (the probe) on the extracted representations. The probe is optimized to distinguish the two domains via binary classification. We report probing accuracy on a held-out test set; an accuracy of $50\%$ (random chance) signifies perfect invariance, whereas higher accuracy indicates significant information leakage.

**Results.** As shown in Table 23, the probe trained on the bias representation $\mathbf{z}_b$ achieves near-perfect accuracy ($96.1\%$), confirming that DIBJUDGE successfully isolates translation artifacts. More importantly, the probe trained on the robust representation $\mathbf{z}_r$ yields an accuracy of approximately $50\%$, demonstrating that predictive features are effectively sanitized of domain-specific signals. In contrast, standard embeddings from the baseline SFT model exhibit substantial leakage ($82.4\%$), highlighting that standard fine-tuning fails to decouple semantic content from stylistic artifacts.

*Table 20.* **Comparison of Heuristic Metrics.** Ablation of the metric used for Bin Classification. NLL performs slightly better, but all metrics provide similar bias mitigation.

| Metric | Bias Severity ↓ |
|---|---|
| Type-Token Ratio (TTR) | 0.17 |
| Perplexity (PPL) | 0.16 |
| Negative Log-Likelihood (NLL) | 0.15 |

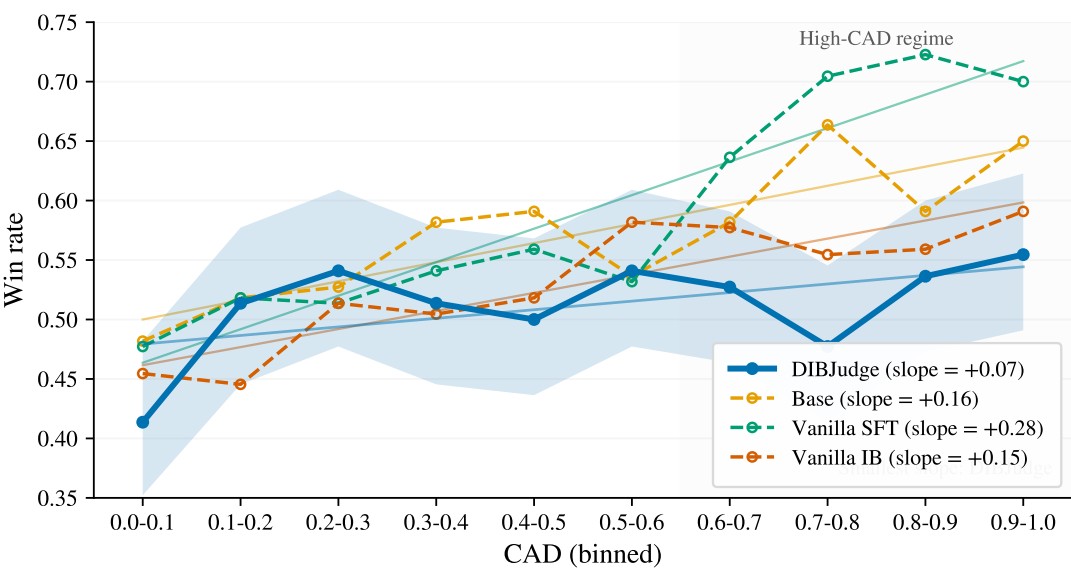

*Figure 7.* **Sensitivity of Win Rate to Cross-Lingual Alignment Discrepancy (CAD).** Win rates of Translationese outputs are plotted against binned CAD values. Baseline judges (dashed lines) display a strong positive dependence on CAD, reflecting an English-anchoring bias toward structurally aligned translations. In contrast, DIBJUDGE (solid line) maintains a near-constant win rate across CAD regimes, including high-CAD regions, indicating robustness to cross-lingual structural divergence.

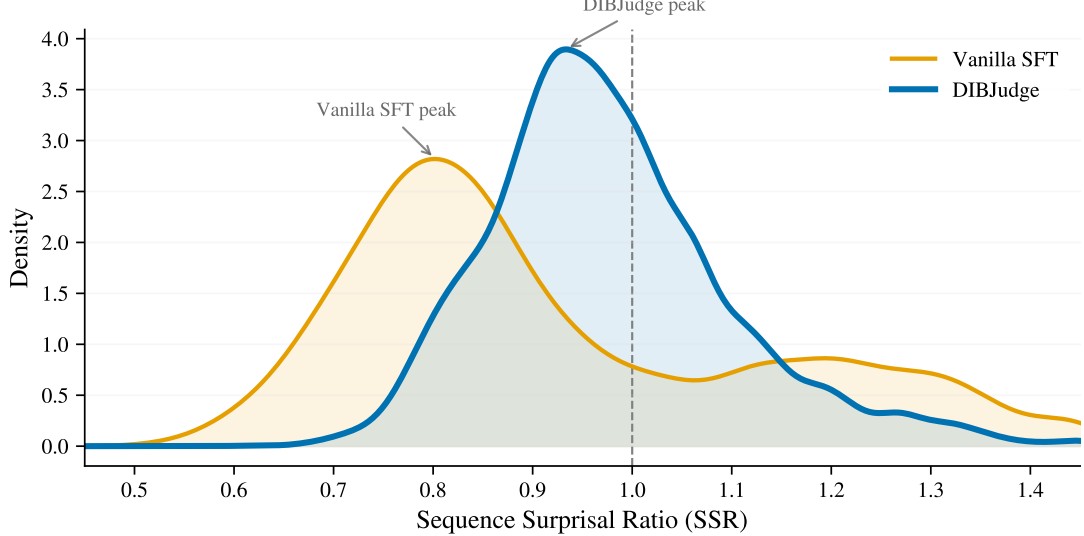

*Figure 8.* **Bias Spectrum Shift via Sequence Surprisal Ratio (SSR).** Kernel density estimates of SSR for instances where the judge selects the machine-generated output (*Machine Wins*). Lower SSR values correspond to spurious correlations with high predictive confidence (i.e., low perplexity), indicating a preference for distributionally simple text. Values near 1 reflect invariance to confidence-related artifacts. While the baseline judge concentrates mass at low SSR values, DIBJUDGE shifts the spectrum toward 1, demonstrating reduced reliance on predictive-confidence shortcuts.

*Table 21.* **Comparison of Bottleneck Mechanisms.** Effect of different latent constraints on *utility* (m-RewardBench accuracy) and *bias mitigation* (Bias Severity). Lower Bias Severity is better.

| Method | Constraint Type | m-RB Acc. ↑ | Bias Sev. ↓ |
|---|---|---|---|
| Low-Rank | Deterministic (Capacity) | 88.2 | 0.185 |
| Noise | Stochastic (Additive) | 85.5 | 0.182 |
| VQ | Discrete (Structural) | 84.9 | **0.082** |
| **Ours** | **Variational (Information)** | **89.3** | **0.091** |

*Table 22.* **Efficiency–bias trade-off of disentanglement mechanisms on Llama-3-8B.** Training latency is normalized to standard fine-tuning ($1.0\times$). Bias Severity is defined in § 2, where lower is better.

| Method | Computational Cost | Latency ↑ | Bias Severity ↓ |
|---|---|---|---|
| Baseline (No disentanglement) | – | $1.00\times$ | 0.284 |
| $\mathcal{L}_{\text{orth}}$ (Orthogonality) | $O(d)$ | **$1.05\times$** | 0.215 |
| MINE (Belghazi et al., 2018) | $O(\text{aux. net})$ | $2.42\times$ | 0.145 |
| CLUB (Cheng et al., 2020) | $O(\text{aux. net})$ | $2.12\times$ | 0.141 |
| HSIC (Gretton et al., 2005) | $O(n^2)$ | $2.78\times$ | **0.138** |
| $\mathcal{L}_{\text{cov}}$ (Ours) | $O(d^2)$ | **$1.18\times$** | 0.142 |

*Table 23.* **Linear probing for domain classification.** Higher accuracy indicates stronger domain information. Effective disentanglement yields *high* accuracy for bias representations and *low* accuracy for robust representations.

| Model | Probed Representation | Accuracy (%) |
|---|---|---|
| Baseline (SFT) | Standard embedding $\mathbf{h}$ | 82.4 |
| **DIBJUDGE (Ours)** | Bias representation $\mathbf{z}_b$ | **96.1** |
|  | Robust representation $\mathbf{z}_r$ | **50.3** |

