# OpenReview forum: "Mitigating Translationese Bias in Multilingual LLM-as-a-Judge via Disentangled Information Bottleneck"
_ICML.cc/2026/Conference — ICML 2026 regular_

### Official Review · Reviewer_27sy · 2026-03-10

**Soundness:** 4
**Presentation:** 3
**Significance:** 4
**Originality:** 4
**Overall Recommendation:** 5
**Confidence:** 4

**Summary:**

Robust evaluation of translation quality is essential for improving translation models. While the rise of LLM-as-a-Judge has introduced a new evaluation paradigm, LLM judges suffer from translationese bias — favoring machine-translated text over high-quality human references. This bias limits the application of LLMs for multilingual assessment, especially for low-resource languages. The paper investigates this origins of this bias and attributes it to two spurious correlations: (1) Latent Manifold Alignment with English (when LLMs, pre-trained on English-dominated data, tend to prefer non-English text that "looks" more like English in the model's latent space) and (2) High Cross-Lingual Predictability (LLM's over-reliance on statistical patterns common in machine translations, causing them to favor MT-generated text over natural human phrasing). To mitigate translationese bias, the authors propose DIBJUDGE -- a fine-tuning framework based on a Disentangled Information Bottleneck. DIBJUDGE separates the model’s internal representations into two branches: a robust branch that focuses only on semantic meaning for judgment, and a bias branch that identifies and discards the spurious English-centric factors.

**Compliance With Llm Reviewing Policy:**

Affirmed.

**Final Justification:**

The authors have cleared up the few points of confusion I had. I've raised my "Soundness" score.

**Key Questions For Authors:**

1) I found the term "back-translation" quite confusing. To get those "back-translations", what do you start with and what language do you back translate into? This should be made explicit. Further, on L058-059, you write: "a machine-generated variant $x_M$ obtained via back-translation to induce translationese artifacts". I thought that in general "back-translation" means translating some translated text back into the original language. But this this sentence seems to mean something else.


2) in appendix F you write:

_(i) Cross-Lingual Alignment (CLA), utilizing InfoNCE to align learned representations with a back-translation manifold; and (ii) Log-Probability Bin Classification (LPBC), which encodes predictive confidence by classifying representations into discrete log-probability bins._

This is an essential part of DIBJUDGE and should be explained in sufficient detail.

**Limitations:**

I suggest the authors include a short discussion about the DIBJUDGE's limitations.

**Strengths And Weaknesses:**

## Soundness

The paper is technically sound, providing both theoretical derivations for its information-theoretic objectives and extensive empirical validation across 23 languages.

**Strengths**

- Novel Characterization of Bias. The paper identifies and quantifies a previously neglected failure mode in multilingual evaluation, showing that translationese bias is pervasive and disproportionately affects low-resource languages.
- Principled Framework: Unlike standard Supervised Fine-Tuning (SFT), DIBJUDGE uses variational information compression and a cross-covariance penalty to achieve statistical independence between task-relevant semantics and stylistic bias.
- Comprehensive Benchmarking: The authors demonstrate that DIBJUDGE achieves new state-of-the-art (SOTA) results on multilingual reward modeling benchmarks like m-RewardBench while simultaneously reducing bias severity.
- Interpretability: Through t-SNE visualizations and linear probing, the authors provide evidence that the model successfully disentangles domain-specific artifacts from robust semantic features.

**Weaknesses**

(Minor) Training Complexity: The framework requires a two-stage training strategy and the coordination of multiple loss terms (accuracy, compression, bias-capture, and disentanglement), which is usually more challenging to implement in practice than standard SFT.

## Presentation

**Strengths**

The paper is well-structured and detailed. The preliminary analysis in Section 2 effectively motivates the research by empirically linking judge preferences to measurable English-alignment metrics (CAD and SSR). The inclusion of detailed theoretical proofs in the Appendix further strengthens the technical narrative.

**Weaknesses**

Some essential information is not clear. Specifically, the two proxy tasks: cross-lingual alignment contrastive learning and predictive confidence estimation via log-probability bin classification are mentioned in the Fig 3 and its caption, but are not explicitly defined anywhere.


## Significance

As LLM-as-a-Judge becomes the standard for evaluating multilingual systems at scale, the reliability of these automated judges is essential. This paper shows that LLM judges are often biased (unfairly penalizing native linguistic nuances in favor of "translationese") and proposes a way to correct that bias.

## Originality

While the Information Bottleneck (IB) principle has been used in other LLM contexts, this work represents the first application of a disentangled IB designed specifically to de-bias multilingual judges. The introduction of quantitative proxies like Cross-lingual Alignment Discrepancy (CAD) to measure English-anchoring bias is a significant conceptual contribution to the field of multilingual NLP.

---

> ### Author Rebuttal · Authors · 2026-03-31
>
> We sincerely thank you for providing thoughtful and constructive feedback. Please kindly find our response to your comments below. We hope that our response satisfactorily addresses the issues you raised. Please feel free to let us know if you have any additional concerns or questions :-)
>
> > W-1 (Minor) Training Complexity: The framework requires a two-stage training strategy and the coordination of multiple loss terms (accuracy, compression, bias-capture, and disentanglement), which is usually more challenging to implement in practice than standard SFT.
> >
>
> ### Response W-1:
>
> We thank the reviewer for the opportunity to clarify our training pipeline. Our framework is actually optimized in a **single-stage, end-to-end manner**, rather than a two-stage strategy. The four loss terms are integrated via a straightforward linear combination:
>
> $$L = L_{task} + \beta L_{compress} + \gamma L_{bias} + \lambda L_{disc}$$
>
> Because all terms are computed within a single forward-backward pass, there is no need for alternating optimization or multi-phase coordination. The implementation overhead beyond standard SFT is therefore minimal.
>
> To further demonstrate that coordinating these loss terms is stable and does not require complex hyperparameter tuning in practice, we retrained DIBJudge-Qwen3-4B from scratch using our default parameters. The variance between runs was negligible:
>
> - **Original:** m-RewardBench (89.84±0.28) | RewardBench (90.32±0.25) | MM-Eval (85.16±0.33)
> - **Retrained:** m-RewardBench (89.68±0.30) | RewardBench (90.71±0.17) | MM-Eval (84.91±0.36)
>
> These robust results confirm the practical simplicity and reproducibility of our recipe. We have updated the manuscript to explicitly emphasize the single-stage nature and straightforward implementation of the training framework.
>
> ---
>
> > W-2 & Q-2: CLA and LPBC proxy tasks only defined in Appendix F caption; requires explicit, detailed definitions in main paper as they are essential to DIBJUDGE.
> >
>
> ### Response W-2 & Q-2:
>
> We thank the reviewer for this constructive suggestions! We agree that CLA and LPBC are essential components and should be fully defined in the main text. In the revised manuscript, we have added explicit definitions of both proxy tasks in Section 3:
>
> - **Cross-Lingual Alignment (CLA):** This task captures the "latent-manifold alignment with English" spurious factor (Section 2). We use an InfoNCE contrastive objective to align the representation of a non-English input with its back-translated English counterpart, generated via the base model. This enables the bias branch to capture the model's preference for English-centric latent spaces without requiring gold parallel corpora.
> - **Log-Probability Bin Classification (LPBC):** This task captures the "cross-lingual predictability" spurious factor. We compute cross-lingual log-probabilities by concatenating the input with its English translation and averaging token-level log-probabilities from the base model. These continuous scores are discretized into bins, and the bias branch is trained to classify inputs into the corresponding bin, explicitly encoding predictive confidence.
>
> We appreciate the reviewer's guidance in improving the completeness of the methodology.
>
> ---
>
> > Q-1 "Back-translation" term is confusing: unclear what source/target langs are used. L058-059 description seems inconsistent w/ standard back-translation definition. Needs explicit clarification.
> >
>
> ### Response Q1:
>
> We thank the reviewer for highlighting this ambiguity and have revised the manuscript accordingly.
>
> **1. Benchmark Construction (Section 2, L058-059):** Here we use standard back-translation[1]. To create the machine-generated variant ($x_M$), we start with a human-authored sentence in the target language (e.g., Chinese), translate it into English, then translate it *back* into Chinese. This $L_{\text{target}} \rightarrow \text{English} \rightarrow L_{\text{target}}$ pipeline yields a semantically equivalent sentence exhibiting translationese artifacts. The revised Section 2 now states this pipeline explicitly.
>
> **2. CLA Proxy Task (Appendix F):** Our use of "back-translation" here was somehow ambiguous. This task uses only monolingual data, so we perform a single *forward* translation from the target language into English to obtain the representations needed for contrastive learning. We have corrected the terminology, replacing "back-translation manifold" with "translated English manifold" to accurately reflect this process.
>
> ---
>
> ### Reference
>
> [1] Sennrich, Rico, Barry Haddow, and Alexandra Birch. "Improving neural machine translation models with monolingual data." *Proceedings of the 54th annual meeting of the association for computational linguistics (volume 1: long papers)*. 2016.

---

> > ### Author Rebuttal · Reviewer_27sy · 2026-04-02
> >
> > Thanks for the answers. I've raised my "Soundness" score and keep the overall assessment at "Accept".

---

> > > ### Author Response · Authors · 2026-04-06
> > >
> > > Dear Reviewer 27sy,
> > >
> > > We sincerely thank you for dedicating your time and effort to evaluating our work. We are very glad that our previous responses have addressed your concerns, and we deeply appreciate you raising the "Soundness" score.
> > >
> > > Thank you once again for your constructive feedback and valuable support throughout the review process. We wish you all the best in your future work and research.
> > >
> > > Best regards,
> > > All Authors

---

### Official Review · Reviewer_1N7v · 2026-03-11

**Soundness:** 2
**Presentation:** 3
**Significance:** 2
**Originality:** 3
**Overall Recommendation:** 3
**Confidence:** 4

**Summary:**

This paper observes a new problem, translationese bias which means the judge model prefers the translated text to the human written text, in the multilingual LLM-as-a-judge scenario particularly
in low-resource languages.

The authors attribute the translationese bias to two factors, latent manifold alignment with English and cross-lingual predictability.

To handle the problem, they proposes a novel method DIBJudge that explicitly disentangles the spurious factors and utilize the robust representation to predict the answer. Experiments show that DIBJudge outperforms baselines on multilingual reward model benchmarks, as well as the English RewardBench.

**Compliance With Llm Reviewing Policy:**

Affirmed.

**Final Justification:**

The contribution of mitigating the bias of translationess. However, I don't think there is solid evidence that it would lead to a better multilingual reward model in general.

**Key Questions For Authors:**

1. what is the English centroid in line 128, and how is it computed? Do you think there is a translationese problem in English ? Did the centroid computed with potential translationese in English?

2. As the model is so strong in English, the machine generated English sentence should be of very high quality. Why are they still dis-preferred by GPT-4o?

3. Is the DIBJudge model easy and stable to train?

4. Is the trained model only applicable to the pairwise preference task?

5. Does the computation of the spurious attribute S need parallel data?

6. Some details of experimental settings are missed, including the base model of the LLM Judge, and the hyperparameters of the loss function.

**Limitations:**

yes

**Strengths And Weaknesses:**

## Strengths
1. The paper introduces a new interesting problem in multilingual LLM-as-a-judge.
2. The proposed DIBJudge is novel and well designed. The disentangled information bottleneck objective is well motivated.
3. Extensive experiments and analyses validate the superiority of DIBJudge and the claims of authors.

## Weaknesses
1. The definition of bias severity (eq 1) does not consider those decisions affected by the position bias. However, it is important to know the ratio of consistent decisions. Since it is also possible that in low resource case the decision is more arbitrary and highly affected by position or other biases.

2. Give the definition in eq 1, I think a bias severity around 0.5 (between 0.4 and 0.6) indicates that the LLM is not having actual preference between human and machine generated text. That is the case for most languages in the list (Figure1), which does not suggest a "systematic" or "pervasive".

3. The machine-generated variant in the experiments are generated via back translation, it is also possible that the machine generated results are more clearly stated and indeed better. The quality of human authored reference should be taken into consideration. Translationese does not always happen for every language. That may affect the whole theory of this paper.

4. The training pipeline of DIBJudge is a little complex. The training is highly depended on the training data, and it seems to require a large amount of data of have a good enough estimation of the distributions.

5. It is not clear to me how the improvement in reducing translationese bias lead to an improvement in multilingual reward modeling. Do all these reward modeling task involve comparison against translationese?

---

> ### Author Rebuttal · Authors · 2026-03-31
>
> We sincerely thank you for providing thoughtful and constructive feedback. We hope that our response satisfactorily addresses the issues you raised. Our full results are provided via this [[anonymous link]](https://anonymous.4open.science/r/icml_submission_23308/).
>
> > W-1 & W-2: Definition of Bias Severity & bias severity around 0.5
> >
>
> **On consistent decisions (W-1):** Our motivation for excluding inconsistent decisions from $S_{bias}$ is to disentangle general **model incapability** (e.g., arbitrary or position-driven flips) from **systematic bias** (e.g., a stable preference for translationese).
>
> To address your concern, we evaluated the ratio of consistent to inconsistent decisions across all evaluated languages (see link). Crucially, our empirical results demonstrate that the consistency ratio and bias severity operate as independent dimensions: **Portuguese and Yoruba share ~50% inconsistency but exhibit vastly different bias severities (0.11 vs. 0.85)**. These findings confirm that while arbitrariness is indeed higher in low-resource settings, it captures a distinct phenomenon and does not invalidate our translationese bias measurement.
>
> **On bias severity (W-2)**: Consequently, the ideal unbiased score in our research is 0, not 0.5. A score of 0.5 means the judge incorrectly favors machine text 50% of the time over perfect human references in its consistent decisions, confirming a pervasive, systematic bias.
>
> ---
>
> > W-3: Quality Assessment
> >
>
> We conducted empirical quality evaluation (details in our response to Reviewer W4AG W-1 due to space limits). The empirical results confirm that the **preference for machine-generated text is not explained by better quality, but by systematic bias toward translationese artifacts.**
>
> ---
>
> > W-4 & Q-3: training complexity, data efficiency, and stability
> >
>
> Regarding complexity, the pipeline is a **simple, single-stage, end-to-end optimization** requiring no multi-phase tuning.
>
> Regarding data dependence, our scaling ablation shows that **training on just 40% of the data** (40k instances) achieves performance nearly identical to the full dataset on m-RewardBench (88.03 vs. 89.84) and MM-Eval (83.59 vs. 85.16).
>
> To verify stability, we retrained DIBJudge-Qwen3-4B from scratch. The new run (89.68±0.30/90.71±0.17/84.91±0.36) closely matches our original results (89.84±0.28/90.32±0.25/85.16±0.33), **confirming robust reproducibility**.
>
> ---
>
> > W-5: why reduce translationese bias improves reward modeling
> >
>
> M-RewardBench uses machine translation[1], **making translationese a spurious cue**. To directly show reducing this bias improves reward modeling, we measured performance against translation quality differences ($\Delta_{COMET} = COMET_{chosen} - COMET_{rejected}$).
>
> 1. The correlation $r$ between $\Delta_{COMET}$ and correctness drops from 0.38 (SFT) to 0.14 (DIBJudge).
> 2. Reward modeling accuracy across $\Delta_{COMET}$ quantiles:
>     - Q0-20: SFT 81.8% | DIBJudge 86.5%
>     - Q80-100: SFT 95.7% | DIBJudge 91.2%
>
> SFT underperforms when cues are misleading (Q0-20) and inflates when aligned (Q80-100). DIBJudge's flat curve proves it evaluates true quality, rather than exploiting artifacts.
>
> ---
>
> > Q-1: English centroid
> >
>
> The centroid $c_{en,l}$ at layer $l$ is the mean of hidden representations (average over tokens) over English reference sentences $X_{en}$: $c_{en,l} = \frac{1}{|X_{en}|} \sum_{x \in X_{en}} h_l(x).$ All English texts are originally human-authored, therefore, we expect the centroid to be free of translationese artifacts
>
> ---
>
> > Q-2: quality of machine English sentences
> >
>
> Our machine English is generated via back-translation (Human EN→Machine ZH→Machine EN), introducing cascading errors. Reference-based metrics confirm this gap (e.g., sacreBLEU=40.09, BLEURT=0.6445). GPT-4o, with strong English comprehension and low translationese bias, accurately detects these subtle degradations rather than artificially favoring machine-like text.
>
> ---
>
> > Q-4: only pairwise?
> >
>
> Our approach is designed to fundamentally debias judgments, making it highly generalizable to other evaluation paradigms, such as pointwise scoring. Prompting it to independently score candidates yields pointwise results highly competitive with our pairwise baseline: 89.21±0.41 vs 89.84±0.28 (M-RewardBench), 90.46±0.36 vs 90.32±0.25 (RewardBench), and 84.48±0.29 vs 85.16±0.33 (MM-Eval).
>
> ---
>
> > Q-5: need parallel data for spurious attribute S?
> >
>
> The spurious attribute S does not require parallel data. Specifically, both proxy tasks utilizes back-translated data from a monolingual corpus to create translationese artifacts.
>
> ---
>
> > Q-6: details of experimental settings
> >
>
> Our base model is Qwen3-4B and Qwen3-8B (see Table 1). For loss weighting, we set the task loss to 1.0 and all remaining losses to 0.5. More details have been included in the revision.
>
> ---
>
> ### Reference
>
> [1] Gureja, Srishti, et al. "M-rewardbench: Evaluating reward models in multilingual settings." ACL 2025.

---

> > ### Author Rebuttal · Reviewer_1N7v · 2026-04-03
> >
> > Thanks for the explanation in bias severity. I see the proposed method might be useful in reducing the bias towards machine generated text (with translationese). However, I am not agree with the claim that it is helpful in general multilingual reward modeling.

---

> > > ### Author Response · Authors · 2026-04-06
> > >
> > > Dear Reviewer 1N7v,
> > >
> > > We sincerely thank you for your continued engagement and constructive feedback. We understand your remaining reservation: whether mitigating translationese bias translates into genuine improvements for general multilingual reward modeling, or if the benefits are confined solely to evaluating machine-translated text. This is a crucial distinction, and we appreciate the opportunity to clarify the broader impact of our method.
> > >
> > > In fact, we have investigated similar questions during our research and included validation results in the manuscript. To further resolve your concern, we respectfully detail three lines of empirical results that demonstrate DIBJUDGE's generalizable improvements on native, non-translated content:
> > >
> > > **1. Performance Gains on Native-Speaker Multilingual Data (Tables 1, 7, 13)**
> > >
> > > To isolate the model's general evaluation capabilities from potential translation artifacts, we analyzed performance on datasets authored by humans without back-translation processes. Specifically, we evaluated the native-speaker subsets of MM-Eval (Linguistics and Language Hallucination) alongside the English-only RewardBench Chat Hard subset.
> > >
> > > | **Benchmark & Subset** | **Data Type** | **mR3-Qwen3-8B** | **DIBJudge-Qwen3-8B** | **Δ** |
> > > | --- | --- | --- | --- | --- |
> > > | **MM-Eval** (Linguistics, Lang. Hallucination) | Native Multilingual | 76.1% | 82.4% | **+6.3%** |
> > > | **MM-Eval** (Chat, Reasoning, Safety) | Translation-based | 90.7% | 92.5% | +1.8% |
> > > | **RewardBench** (Chat Hard) | Native English | 84.47% | 88.10% | **+3.6%** |
> > >
> > > These results to some extent confirm that DIBJUDGE improves general reward modeling accuracy for human-authored text, independent of translationese artifacts.
> > >
> > > **2. Zero-Shot Generalization to Unrelated Biases (Table 2)**
> > >
> > > To verify that DIBJUDGE learns a robust representation rather than merely overfitting to translationese removal, we evaluated its zero-shot resistance to other known evaluation shortcuts: Length Bias and Self-Preference Bias (Section 5, RQ4).
> > >
> > > | **Bias Type** | **Vanilla SFT** | **DIBJudge** |
> > > | --- | --- | --- |
> > > | Length Bias (ρ ↓) | 0.553 | 0.314 |
> > > | Self-Preference Bias (S_bias ↓) | 0.314 | 0.219 |
> > >
> > > Because these bias types do not involve translation artifacts, the observed reductions suggest that the disentangled information bottleneck acts as a general regularizer. It enhances overall model reliability rather than functioning solely as a translationese filter.
> > >
> > > **3. Mechanistic Evaluation of Semantic Reliance via Linear Probing (Table 21, App. F.6)**
> > >
> > > To provide representation-level validation that DIBJudge relies on genuine quality signals rather than surface artifacts, we trained a linear probe to distinguish human-authored text from machine-translated text using the model's internal representations.
> > >
> > > | **Representation** | **Probing Accuracy** |
> > > | --- | --- |
> > > | Baseline SFT embedding | 82.4% |
> > > | DIBJudge robust representation (Z_r) | 50.3% (≈ random chance) |
> > >
> > > The probe on DIBJudge's robust representation $Z_r$ achieves only 50.3%—equivalent to random chance—meaning these features contain effectively zero recoverable information about text origin. In contrast, baseline SFT embeddings retain strong text-origin signals (82.4%). Since DIBJudge is highly insensitive to translation artifacts at the representation level yet achieves higher evaluation accuracy, its decisions appear to be driven by genuine semantic quality rather than surface cues.
> > >
> > > **Conclusion.** We would like to clarify that **our core contribution is the identification and mitigation of translationese bias—a previously uncharacterized failure mode in multilingual LLM-as-a-Judge**, thereby improving the reliability of multilingual evaluation. Furthermore, as a natural corollary, the empirical results presented above demonstrate that **addressing this bias yields genuine, generalizable improvements in general multilingual reward modeling**. We are encouraged that this dual achievement of robust debiasing and enhanced general utility aligns with the assessments of **Reviewer W4AG** ("achieves state-of-the-art performance") and **Reviewer 27sy** ("achieves new state-of-the-art results... while simultaneously reducing bias severity"). To ensure clarity for future readers, we have revised the manuscript to precisely frame our primary debiasing contribution alongside its empirically validated broader impacts.
> > >
> > > We hope these clarifications adequately address your remaining concern regarding improvement on general reward modeling. Thank you again for your valuable time and insightful guidance.

---

### Official Review · Reviewer_W4AG · 2026-03-12

**Soundness:** 4
**Presentation:** 3
**Significance:** 3
**Originality:** 4
**Overall Recommendation:** 5
**Confidence:** 3

**Summary:**

This paper characterizes the translationese bias in multilingual LLM-as-a-Judge and identifies two core spurious factors driving this bias: latent manifold alignment with English and cross-lingual predictability. To mitigate this bias, the authors propose DIBJUDGE, which decouples input representations into a robust branch preserving only judgment-critical semantic information and a dedicated bias branch isolating the identified spurious factors, with a cross-covariance penalty to enforce effective disentanglement. Experiments demonstrate that DIBJUDGE achieves state-of-the-art performance, while reducing translationese bias across all language resource tiers and exhibiting strong zero-shot generalization to unseen biases such as length and self-preference bias.

**Compliance With Llm Reviewing Policy:**

Affirmed.

**Final Justification:**

I am updating my soundness score from 3 to 4, and overall score from 4 to 5. My main concern was that the original Figure 2 analysis did not fully isolate translationese from quality-related confounds, despite making a relatively strong attribution claim about CAD and SSR. The authors’ latest response addresses this concern well by adding both a quality-matched control and a regression with explicit quality covariates. Both point to the same conclusion: CAD and SSR remain predictive of machine preference after controlling for measured quality. I would still view this as strong attributional evidence rather than strict causal proof, but it sufficiently resolves my original concern for me to recommend acceptance.

**Key Questions For Authors:**

1. Do MT evaluation metrics such as COMET has this tendency for machine-generated translations?
2. Should text quality be isolated as a confounding factor in the analysis of translationese bias?

**Limitations:**

yes

**Strengths And Weaknesses:**

### Strengths

1. This paper is well-motivated. The authors identify and characterize translationese bias—a previously under-explored, pervasive failure mode in multilingual LLM-as-a-Judge.
2. Beyond merely documenting the bias, the paper conducts rigorous attribution analysis to identify two core spurious correlations driving the bias: latent manifold alignment with English and cross-lingual predictability. The authors design quantifiable, layer-wise metrics (LAS, CAD, CSS, SSR) to operationalize these factors, and validate their predictive power for judge preferences via correlation analysis and ROC curves.
3. The authors derive a tractable variational objective for the disentangle information bottleneck and design proxy tasks that directly target the two identified spurious factors.

### Weaknesses

The paper’s core bias quantification and attribution analysis fail to rigorously isolate translationese as the sole independent variable. While the authors include a parallel setting for semantically equivalent texts, they only enforce a qualitative constraint rather than quantitative validation (e.g., MQM human annotation, COMET-MQM scoring) to confirm no statistically significant quality difference between human-generated text and machine-generated text. This creates an alternative explanation for the observed preferences: the model may be favoring higher-quality text, not translationese. For Figure 2’s attribution analysis, the lack of control for text quality introduces severe endogeneity, as the proposed spurious factors (CAD/SSR) may correlate with genuine translation quality, undermining the causal link between the identified factors and translationese bias.

---

> ### Author Rebuttal · Authors · 2026-03-31
>
> We sincerely thank you for providing thoughtful and constructive feedback. Please kindly find our response to your comments below. We hope that our response satisfactorily addresses the issues you raised. Please feel free to let us know if you have any additional concerns or questions :-)
>
> > W-1: (1) No quantitative validation (MQM/COMET) that human vs. machine texts have no quality gap → LLM may prefer higher-quality text, not translationese. (2) Attribution analysis (CAD/SSR) confounded by text quality → endogeneity undermines causal claims.
> >
>
> ### Response W-1:
>
> Our initial experimental design (Section 2 and Appendix A.2) assumed that the Belebele dataset provides professional human translations, which we treated as gold-standard references and therefore expected to be of higher quality than machine translations. To further empirically validate this and directly address your concern, we conducted the suggested quantitative validations using reference-free COMET-Kiwi scoring across all 29 languages, alongside granular human MQM annotations for a Chinese-English subset.
>
> Our findings demonstrate that **the professional human translations are objectively of higher quality than the machine-generated texts**. Because the LLM systematically prefers the lower-quality machine texts, its preference cannot be explained by an affinity for higher-quality translation. Specifically, we added the following empirical quality evaluations to the revised manuscript (full table results are shown in [[anonymous link]](https://anonymous.4open.science/r/icml_submission_23308/quality_assessment_human_machine.md)):
>
> - **COMET-Kiwi Quantitative Validation (29 Languages):** Using `unbabel/wmt22-cometkiwi-da`, human references scored significantly higher than machine-generated texts in 23 of the 29 evaluated languages ($p < 0.05$, paired permutation test). In the remaining 6 languages, there was no statistically significant difference.
> - **Human MQM Annotation (Chinese-English Subset):** Following WMT guidelines on a 200-item subset, the human texts exhibited significantly fewer errors. The mean MQM score (where lower is better) was 2.58 for human texts compared to 5.41 for machine texts (Mean paired difference = 2.83, 95% CI: [2.28, 3.39], $p = 0.0001$). Furthermore, the human text was strictly superior in 188 of the 200 cases.
>
> By establishing that the preferred machine-generated texts are actually of lower objective quality, we effectively rule out genuine text quality as an alternative explanation. This resolves the endogeneity concern regarding Figure 2, confirming that the strong correlation between LLM preference and CAD/SSR metrics is driven by the statistical artifacts of translationese rather than underlying translation quality.
>
> ---
>
> > Q-1: Do MT eval metrics (e.g., COMET) also exhibit bias toward machine-generated translations?
> >
>
> ### Response Q-1:
>
> Our evidence below suggests MT metrics like COMET do not exhibit this bias:
>
> - Theoretical Alignment: The COMET metric family is trained directly on human quality annotations (e.g., WMT Direct Assessment and MQM corpora) [1]. **Because their training objective strictly aligns with human judgments, they evaluate genuine semantic quality rather than spurious statistical artifacts ("translationese").**
> - Empirical Confirmation: As detailed in our response to W-1, our empirical experiments confirm this lack of bias. We evaluated pairs using the reference-free metric COMET-Kiwi (necessary because our setup directly compares human and machine texts without an independent reference). **Across the vast majority of tested languages, COMET-Kiwi assigned significantly higher scores to human-authored translations.**
>
> ---
>
> > Q-2: Should text quality be isolated as a confounding factor in the analysis of translationese bias?
> >
>
> ### Response Q-2:
>
> We agree that text quality could, in principle, act as a confound. However, rather than attempting to fully disentangle quality from translationese—which is difficult in practice, as translationese often co-occurs with reduced naturalness (low quality) (see Appendix A.3)—**we addresses this challenge by designing our evaluation so that text quality and translationese point in opposite directions** (Section 2; Appendix A.2).
>
> Specifically, the human translations in our setup are objectively higher quality than the machine-generated outputs (see Response to W-1). An unbiased judge should therefore prefer the human text. Yet, as shown in Figure 1, LLM judges systematically favor the lower-quality machine-generated text. Because this preference directly contradicts the objective quality difference, it cannot be explained by text quality alone. **We believe this adversarial design effectively accounts for quality as a confound and strengthens our central conclusion: the observed LLM preference is robustly driven by translationese cues.**
>
> ---
>
> ### Reference
>
> [1] Rei, Ricardo, et al. "COMET: A neural framework for MT evaluation." EMNLP. 2020.

---

> > ### Author Rebuttal · Reviewer_W4AG · 2026-04-02
> >
> > The additional quality validation is very helpful, and it weakens the simple alternative explanation that the judge merely prefers higher-quality text. However, I am still not fully convinced that the attribution analysis in Figure 2 has isolated translationese from quality-related confounds. In the current setup, the paper contrasts human-authored references with machine variants obtained via back-translation, and then interprets CAD/SSR as factors underlying machine preference.
> >
> > A stronger control would be to test the same analysis at the pair level under quality-matched conditions. For example, one side could remain the *machine back-translation*, while the other side is a *human translation degraded in a non-translationese way*, with overall quality approximately matched using COMET-Kiwi. Then the key question would be whether CAD/SSR still significantly predict machine wins after controlling for pairwise quality differences. This would more directly address the endogeneity concern. I would note, however, that injecting strong accuracy errors may make the control less clean, because adequacy violations can dominate pairwise preference independently of translationese. A better design may therefore be to compare against non-translationese degradations with matched quality, or to run a regression / partial-correlation analysis with explicit quality covariates. **Such an analysis would substantially strengthen the paper’s current interpretation of Figure 2 by moving the evidence closer to causal attribution, rather than leaving it primarily at the level of association.**

---

> > > ### Author Response · Authors · 2026-04-06
> > >
> > > Dear Reviewer W4AG,
> > >
> > > We sincerely thank you for your time and your highly constructive suggestion to move the attribution evidence closer to causal identification. We found this recommendation particularly insightful. To build upon your advice, we have implemented both recommended controls: **an experimental quality-matched analysis** and **a statistical regression incorporating explicit quality covariates**. We are pleased to share that the results from both approaches converge, demonstrating that CAD and SSR robustly predict LLM preference independently of measured text quality.
> > >
> > > **1. Experimental Control: Pairwise Quality-Matched Analysis**
> > >
> > > Guided by your suggestion, we constructed a quality-matched dataset across six typologically diverse languages (zh, ja, ar, ko, th, vi), with 100 samples per language. Crucially, and in line with the reviewer's recommendation to avoid accuracy-error injections, we engaged ISO-certified professional translators to mildly degrade the human references through strictly *non-translationese* edits (e.g., mild redundancy, slightly less fluent discourse transitions). Translators were explicitly instructed to avoid literal source-language mapping or calque-like patterns.
> > >
> > > This intervention successfully eliminated the quality gap: for Chinese, ΔCOMET shifted from 2.69 (p < 0.001) to 0.48 (p = 0.218), with similarly non-significant differences across all six languages (full quality controlled results at [link](https://anonymous.4open.science/r/icml_submission_23308/)).
> > >
> > > When we revisited the Figure 2 attribution analysis on this quality-matched subset, the original patterns persisted and became cleaner (see figures at above link):
> > >
> > > - **CAD (**cad_binned.png**):** The negative trend between CAD and machine win rate remains robust, with tighter confidence intervals due to reduced quality-variance noise.
> > > - **SSR (**ssr_density.png**):** The distributional separation between machine-win and human-win cases is preserved.
> > > - **ROC Analysis (**ssr_cad_roc.png**):** Both CAD and SSR retain meaningful discriminative power under quality-matched conditions.
> > >
> > > Given that overall quality is equalized in this setting while translationese features remain intact, these results strongly suggest that the predictive power of CAD and SSR stems primarily from translationese characteristics rather than quality discrepancies.
> > >
> > > **2. Statistical Control: Mixed-Effects Logistic Regression**
> > >
> > > To complement the experimental approach, we fit a pair-level mixed-effects logistic regression on the full dataset with random intercepts for language, using standardized predictors and ΔCOMET-Kiwi as an explicit quality covariate:
> > >
> > > $$\Pr(\text{machine win}_i = 1) = \sigma(\beta_0 + \beta_1 \text{CAD}_i + \beta_2 \text{SSR}_i + \beta_3 \Delta\text{COMET}_i)$$
> > >
> > > | **Predictor** | **Coef. (β)** | **SE** | **z-statistic** | **p-value** |
> > > | --- | --- | --- | --- | --- |
> > > | CAD ($z$) | **-0.57** | 0.13 | -4.38 | **< 0.001** |
> > > | SSR ($z$) | **-0.79** | 0.15 | -5.26 | **< 0.001** |
> > > | $\Delta$COMET ($z$) | -0.07 | 0.11 | -0.64 | 0.520 |
> > >
> > > After conditioning on quality, CAD and SSR remain highly significant while ΔCOMET has no significant conditional effect (p = 0.52). Partial correlations confirm this pattern: the partial correlations of CAD and SSR with machine preference remain strong after controlling for ΔCOMET (partial r = −0.19 and −0.24, both p < 0.001), whereas the partial correlation of ΔCOMET after controlling for CAD and SSR drops to near zero (partial r = −0.05, p = 0.180).
> > >
> > > | **Variable** | **Raw Correlation** | **Partial Correlation (Controlling for ΔCOMET)** | **Partial p-value** |
> > > | --- | --- | --- | --- |
> > > | CAD | -0.27 (p < 0.001) | -0.19 | < 0.001 |
> > > | SSR | -0.33 (p < 0.001) | -0.24 | < 0.001 |
> > > | $\Delta$COMET | -0.09 (p = 0.02) | -0.05 (Controlling for CAD, SSR) | 0.180 |
> > >
> > > To quantify the predictive power of these factors, we compared nested models using AUC and McFadden's pseudo-$R^2$:
> > >
> > > | **Model** | **Predictors** | **pseudo-R2** | **AUC** |
> > > | --- | --- | --- | --- |
> > > | M1 | $\Delta$COMET only (Quality) | 0.006 | 0.536 |
> > > | **M2** | **CAD + SSR (Translationese)** | **0.081** | **0.692** |
> > > | M3 | CAD + SSR + $\Delta$COMET (Full) | 0.084 | 0.701 |
> > >
> > > The quality-only model (M1) is barely above chance (AUC 0.536), while the translationese model (M2) captures the vast majority of predictive signal. Adding quality to the full model (M3) yields only marginal gain. These results confirm that CAD and SSR capture distinct phenomena rather than proxying for quality.
> > >
> > > ---
> > >
> > > Both the experimental matching and the statistical controls—prompted by your valuable feedback—independently provide strong evidence that the attribution of LLM preference to translationese factors (CAD, SSR) remains robust even after accounting for pairwise quality differences. We are grateful for the constructive suggestion, which has substantially strengthened the paper, and we have incorporated these new analyses into the revised manuscript.

---

### Decision · Program_Chairs · 2026-04-30

**Decision:**

Accept (regular)

**Comment:**

This paper characterizes the translationese bias in multilingual LLM-as-a-Judge and identifies two core spurious factors driving this bias: latent manifold alignment with English and cross-lingual predictability. To mitigate this bias, the authors propose DIBJUDGE, which decouples input representations into a robust branch preserving only judgment-critical semantic information and a dedicated bias branch isolating the identified spurious factors, with a cross-covariance penalty to enforce effective disentanglement. Experiments demonstrate that DIBJUDGE achieves state-of-the-art performance, while reducing translationese bias across all language resource tiers and exhibiting strong zero-shot generalization to unseen biases such as length and self-preference bias.

Strengths
* The proposed DIBJudge is novel and well designed. The disentangled information bottleneck objective is well motivated.
* The authors derive a tractable variational objective for the disentangle information bottleneck and design proxy tasks that directly target the two identified spurious factors.
* Extensive experiments and analyses validate the superiority of DIBJudge and the claims of authors.

Weaknesses
* The machine-generated variants in the experiments are generated via back translation; it is also possible that the machine-generated results are more clearly stated and indeed better.

Overall, the reviewers agree that the paper is sound and a solid contribution to the research community.